# A Lie Group Approach to Riemannian Batch Normalization

**Ziheng Chen**[1], **Yue Song**[1]*, **Yunmei Liu**[2] **& Nicu Sebe**[1]
[1] University of Trento, [2] University of Louisville
`ziheng_ch@163.com, yue.song@unitn.it`

## Abstract

Manifold-valued measurements exist in numerous applications within computer vision and machine learning. Recent studies have extended Deep Neural Networks (DNNs) to manifolds, and concomitantly, normalization techniques have also been adapted to several manifolds, referred to as Riemannian normalization. Nonetheless, most of the existing Riemannian normalization methods have been derived in an *ad hoc* manner and only apply to specific manifolds. This paper establishes a unified framework for Riemannian Batch Normalization (RBN) techniques on Lie groups. Our framework offers the theoretical guarantee of controlling both the Riemannian mean and variance. Empirically, we focus on Symmetric Positive Definite (SPD) manifolds, which possess three distinct types of Lie group structures. Using the deformation concept, we generalize the existing Lie groups on SPD manifolds into three families of parameterized Lie groups. Specific normalization layers induced by these Lie groups are then proposed for SPD neural networks. We demonstrate the effectiveness of our approach through three sets of experiments: radar recognition, human action recognition, and electroencephalography (EEG) classification. The code is available at `https://github.com/GitZH-Chen/LieBN.git`.

## 1 Introduction

Over the past decade or so, Deep Neural Networks (DNNs) have achieved remarkable progress across various scientific fields (Hochreiter & Schmidhuber, 1997; Krizhevsky et al., 2012; He et al., 2016; Vaswani et al., 2017). Conventionally, DNNs have been developed with the underlying assumption of the Euclidean geometry inherent to input data. Nonetheless, there exists a plethora of applications wherein the latent spaces are defined by non-Euclidean structures such as manifolds (Bronstein et al., 2017). To address this issue, researchers have attempted to extend various types of DNNs to manifolds based on the theories of Riemannian geometry (Huang & Van Gool, 2017; Huang et al., 2017; 2018; Ganea et al., 2018; Chakraborty et al., 2018; Brooks et al., 2019a;e;c;d; Brooks, 2020; Brooks et al., 2020; Chen et al., 2020; Chakraborty et al., 2020; Chakraborty, 2020; Chen et al., 2021; Wang et al., 2022b;a; Nguyen, 2022a;b; Nguyen & Yang, 2023; Chen et al., 2023c;e;b; Wang et al., 2024).

Motivated by the great success of normalization techniques within DNNs (Ioffe & Szegedy, 2015; Ba et al., 2016; Ulyanov et al., 2016; Wu & He, 2018; Chen et al., 2023a), researchers have sought to devise normalization layers tailored for manifold-valued data. Brooks et al. (2019b) introduced Riemannian Batch Normalization (RBN) specifically designed for SPD manifolds, with the ability to regulate the Riemannian mean. This approach was further refined in Kobler et al. (2022b) to extend the control over the Riemannian variance. *However, the above methods are constrained within the affine-invariant metric (AIM) on SPD manifolds, limiting their applicability and generality.* On the other hand, Chakraborty (2020) proposed two distinct Riemannian normalization frameworks, one tailored for Riemannian homogeneous spaces and the other catering to matrix Lie groups. *Nonetheless, the normalization designed for Riemannian homogeneous spaces cannot regulate mean nor variance, while the normalization approach for matrix Lie groups is confined to a specific type of distance (Chakraborty, 2020, Sec. 3.2).* A principled Riemannian normalization framework capable of controlling both Riemannian mean and variance remains unexplored.

---

*Corresponding author

Given that Batch Normalization (BN) (Ioffe & Szegedy, 2015) serves as the foundational proto-type for various types of normalization, our paper only concentrates on RBN currently and can be readily extended to other normalization techniques. As several manifold-valued measurements form Lie groups, including SPD manifolds, Special Orthogonal (SO) groups, and Special Euclidean (SE) groups, we further direct our attention towards Lie groups. We propose a general framework for RBN over Lie groups, referred to as LieBN, and validate our approach in normalizing both the Riemannian mean and variance. On the empirical side, we focus on SPD manifolds, where three distinct types of Lie groups have been recognized in the literature. We generalize these existing Lie groups into parameterized forms through the deformation concept. Then, we showcase our LieBN framework on SPD manifolds under these Lie groups and propose specific normalization layers. Extensive experiments conducted on widely-used SPD benchmarks demonstrate the effectiveness of our framework. We highlight that our work is entirely theoretically different from Brooks et al. (2019b); Kobler et al. (2022a); Lou et al. (2020), and more general than Chakraborty (2020). The previous RBN methods are either designed for a specific manifold or metric (Brooks et al., 2019b; Kobler et al., 2022a; Chakraborty, 2020), or fail to control mean and variance (Lou et al., 2020), while our LieBN guarantees the normalization of mean and variance on general Lie groups. In sum-mary, our main contributions are as follows: **(a)** a general Lie group batch normalization framework with controllable first- and second-order statistical moments, and **(b)** the specific construction of our LieBN layers on SPD manifolds based on three deformed Lie groups and their application on SPD neural networks. Due to page limites, all the proofs are placed in App. I.

## 2 PRELIMINARIES

This section provides a brief review of the Lie group and the geometries of SPD manifolds. For more in-depth discussions, please refer to Tu (2011); Do Carmo & Flaherty Francis (1992).

**Definition 2.1** (Lie Groups). A manifold is a Lie group, if it forms a group with a group operation $\odot$ such that $m(x, y) \mapsto x \odot y$ and $i(x) \mapsto x_\odot^{-1}$ are both smooth, where $x_\odot^{-1}$ is the group inverse.

**Definition 2.2** (Left-invariance). A Riemannian metric $g$ over a Lie group $\{G, \odot\}$ is left-invariant, if for any $x, y \in G$ and $V_1, V_2 \in T_x\mathcal{M}$,

$$g_y(V_1, V_2) = g_{L_x(y)}(L_{x*,y}(V_1), L_{x*,y}(V_2)), \tag{1}$$

where $L_x(y) = x \odot y$ is the left translation by $x$, and $L_{x*,y}$ is the differential map of $L_x$ at $y$.

A Lie group is a group and also a manifold. The most natural Riemannian metric on a Lie group is the left-invariant metric[1]. Similarly, one can define the right-invariant metric as Def. 2.2. A bi-invariant Riemannian metric is the one with both left and right invariance. Given the analogous properties of left and right-invariant metrics, this paper focuses on left-invariant metrics.

The idea of pullback is ubiquitous in differential geometry and can be considered as a natural coun-terpart of bijection in the set theory.

**Definition 2.3** (Pullback Metrics). Suppose $\mathcal{M}_1, \mathcal{M}_2$ are smooth manifolds, $g$ is a Riemannian metric on $\mathcal{M}_2$, and $f : \mathcal{M}_1 \to \mathcal{M}_2$ is smooth. Then the pullback of $g$ by $f$ is defined point-wisely,

$$(f^*g)_p(V, W) = g_{f(p)}(f_{*,p}(V), f_{*,p}(W)), \tag{2}$$

where $p \in \mathcal{M}$, $f_{*,p}(\cdot)$ is the differential map of $f$ at $p$, and $V, W \in T_p\mathcal{M}$. If $f^*g$ is positive definite, it is a Riemannian metric on $\mathcal{M}_1$, called the pullback metric defined by $f$.

The most common pullback metric is obtained through the pullback of a diffeomorphism $f$. Ad-ditionally, if $\{\mathcal{M}_2, \odot_2\}$ constitutes a Lie group, the diffeomorphism $f$ can pull back the group operation $\odot_2$ to $\odot_1$ on $\mathcal{M}_1$, which is defined as $\forall P, Q \in \mathcal{M}_1, P \odot_1 Q = f^{-1}(f(P) \odot_2 f(Q))$. Henceforth, $\{\mathcal{M}, \odot, g\}$, abbreviated as $\mathcal{M}$, always signify a Lie group with left-invariant metric.

Now, we briefly review the geometry of SPD manifolds. We denote $n \times n$ SPD matrices as $\mathcal{S}_{++}^n$ and $n \times n$ real symmetric matrices as $\mathcal{S}^n$. As shown in Arsigny et al. (2005), $\mathcal{S}_{++}^n$ forms a man-ifold, known as the SPD manifold, and $\mathcal{S}^n$ is a Euclidean space. SPD manifolds exhibit three Lie group structures, each associated with an invariant metric. These metrics include the Log-Euclidean

---

[1] Left invariant metric always exists for every Lie group (Do Carmo & Flaherty Francis, 1992).

Metric (LEM) (Arsigny et al., 2005), Affine-Invariant Metric (AIM) (Pennec et al., 2006), and Log-Cholesky Metric (LCM) (Lin, 2019). In Thanwerdas & Pennec (2023), LEM and AIM are generalized into two-parameter families of metrics, denoted as $(\alpha, \beta)$-LEM and $(\alpha, \beta)$-AIM, respectively. Both $(\alpha, \beta)$-LEM and $(\alpha, \beta)$-AIM are defined by the O($n$)-invariant inner product on the tangent space at the identity matrix, expressed as:

$$\langle V, W \rangle^{(\alpha,\beta)} = \alpha \langle V, W \rangle + \beta \operatorname{tr}(V) \operatorname{tr}(W), \tag{3}$$

where $\langle \cdot, \cdot \rangle$ is the Frobenius inner product, $V, W \in T_I \mathcal{S}_{++}^n \cong \mathcal{S}^n$, $(\alpha, \beta) \in \mathbf{ST} = \{(\alpha, \beta) \in \mathbb{R}^2 \mid \min(\alpha, \alpha + n\beta) > 0\}$.

In fact, $(\alpha, \beta)$-LEM, $(\alpha, \beta)$-AIM, and LCM are all pullback metrics. Specifically, $(\alpha, \beta)$-LEM is the pullback metric from $\mathcal{S}^n$ (Chen et al., 2023c), while $(\alpha, \beta)$-AIM is the pullback metric from a left-invariant metric on the Cholesky manifold (Thanwerdas & Pennec, 2022b). Additionally, as shown in Chen et al. (2023d), LCM is the pullback metric from the Euclidean space $\mathcal{L}^n$ of lower triangular matrices by the diffeomorphism defined as

$$\psi_{\mathrm{LC}}(P) = \lfloor L \rfloor + \mathrm{Dlog}(L), \tag{4}$$

where $P = LL^\top$ represents the Cholesky decomposition, $\lfloor \cdot \rfloor$ is the strictly lower part of a square matrix, and $\mathrm{Dlog}(L)$ is a diagonal matrix consisting of the logarithm of the diagonal element of $L$.

Let $\{w_{1\ldots N}\}$ be weights satisfying a convexity constraint, i.e., $\forall i, w_i > 0$ and $\sum_i w_i = 1$. The weighted Fréchet mean (WFM) of a set of SPD matrices $\{P_{i\ldots N}\}$ is defined as

$$\mathrm{WFM}(\{w_i\}, \{P_i\}) = \operatorname*{argmin}_{S \in \mathcal{S}_{++}^n} \sum_{i=1}^N w_i \, \mathrm{d}^2 (P_i, S), \tag{5}$$

where $\mathrm{d}(\cdot, \cdot)$ denotes the geodesic distance. When $w_i = 1/N$ for all $i$, then Eq. (5) is reduced to the Fréchet mean, denoted as $\mathrm{FM}(\{P_i\})$. The Fréchet variance $v^2$ is the attained value at the minimizer of the Fréchet mean. In this paper, we will interchangeably use the terms Riemannian mean with Fréchet mean, and Riemannian variance with Fréchet variance. For $(\alpha, \beta)$-LEM and LCM, the Fréchet mean has a closed-form expression. Moreover, since $(\alpha, \beta)$-AIM has non-positive sectional curvature (Thanwerdas & Pennec, 2023, Tab. 5), the Fréchet mean exists uniquely (Berger, 2003, 6.1.5) and can be computed by the Karcher flow algorithm (Karcher, 1977).

Given SPD matrices $P, Q \in \mathcal{S}_{++}^n$ along with tangent vectors $V, W \in T_P \mathcal{S}_{++}^n$, we introduce the following notations. Specifically, we denote $g_P(\cdot, \cdot)$ as the Riemannian metric at $P$, $\mathrm{Log}_P(\cdot)$ as the Riemannian logarithm at $P$, $\mathrm{mexp}(\cdot)$ and $\mathrm{mlog}(\cdot)$ as the matrix exponentiation and logarithm, and $\mathrm{d}(\cdot, \cdot)$ as the geodesic distance, respectively. Additionally, $\mathrm{Chol}(\cdot)$ signifies the Cholesky decomposition. We use $L = \mathrm{Chol}(P)$ and $K = \mathrm{Chol}(Q)$ to denote the Cholesky factor of $P$ and $Q$. $\mathbb{K}$ and $\mathbb{L}$ are diagonal matrices with diagonal elements from $K$ and $L$. $\mathrm{mlog}_{*,P}$ and $(\mathrm{Chol})_{*,L}^{-1}$ represent the differentials of $\mathrm{mlog}$ and $\mathrm{Chol}^{-1}$ at $P$ and $L$. We denote $\| \cdot \|^{(\alpha,\beta)}$ and $\| \cdot \|_\mathrm{F}$ as the norm induced by $\langle \cdot, \cdot \rangle$ and the standard Frobenius norm. We summarize all the necessary ingredients in Tab. 1.

Table 1: Lie group structures and the associated Riemannian operators on SPD manifolds.

| Metric | $(\alpha, \beta)$-LEM | $(\alpha, \beta)$-AIM | LCM |
|---|---|---|---|
| $g_P(V, W)$ | $\langle \mathrm{mlog}_{*,P}(V), \mathrm{mlog}_{*,P}(W) \rangle^{(\alpha,\beta)}$ | $\langle P^{-1}V, WP^{-1} \rangle^{(\alpha,\beta)}$ | $\sum_{i>j} V_{ij}W_{ij} + \sum_{j=1}^n V_{jj}W_{jj}L_{jj}^{-2}$ |
| $\mathrm{d}(P, Q)$ | $\| \mathrm{mlog}(P) - \mathrm{mlog}(Q) \|^{(\alpha,\beta)}$ | $\left\| \mathrm{mlog}\left( Q^{-\frac{1}{2}}PQ^{-\frac{1}{2}} \right) \right\|^{(\alpha,\beta)}$ | $\| \psi_{\mathrm{LC}} \circ \mathrm{Chol}(P) - \psi_{\mathrm{LC}} \circ \mathrm{Chol}(Q) \|_\mathrm{F}$ |
| $Q \odot P$ | $\mathrm{mexp}(\mathrm{mlog}(P) + \mathrm{mlog}(Q))$ | $KPK^\top$ | $\mathrm{Chol}^{-1}(\lfloor L + K \rfloor + \mathbb{K}\mathbb{L})$ |
| $\mathrm{FM}(\{P_i\})$ | $\mathrm{mexp}\left( \frac{1}{n} \sum_i \mathrm{mlog} P_i \right)$ | Karcher Flow | $\psi_{\mathrm{LC}}^{-1}\left( \frac{1}{n} \sum_i \psi_{\mathrm{LC}}(P_i) \right)$ |
| $\mathrm{Log}_P Q$ | $(\mathrm{mlog}_{*,P})^{-1}\left[\mathrm{mlog}(Q) - \mathrm{mlog}(P)\right]$ | $P^{\frac{1}{2}} \mathrm{mlog}\left( P^{\frac{-1}{2}}QP^{\frac{-1}{2}} \right) P^{\frac{1}{2}}$ | $(\mathrm{Chol}^{-1})_{*,L}\left[ \lfloor K \rfloor - \lfloor L \rfloor + \mathbb{L} \, \mathrm{Dlog}(\mathbb{L}^{-1}\mathbb{K}) \right]$ |
| Invariance | Bi-invariance | Left-invariance | Bi-invariance |

## 3 REVISITING NORMALIZATION

### 3.1 REVISITING EUCLIDEAN NORMALIZATION

In Euclidean DNNs, normalization stands as a pivotal technique for accelerating network training by mitigating the issue of internal covariate shift (Ioffe & Szegedy, 2015). While various normalization

methods have been introduced (Ioffe & Szegedy, 2015; Ba et al., 2016; Ulyanov et al., 2016; Wu & He, 2018), they all share a common fundamental concept: the regulation of the first and second statistical moments. In this paper, we focus on batch normalization only.

Given a batch of activations $\{x_{i...N}\}$, the core operation of batch normalization can be expressed as:

$$\forall i \leq N, x_i \leftarrow \gamma \frac{x_i - \mu_b}{\sqrt{v_b^2 + \epsilon}} + \beta \tag{6}$$

where $\mu_b$ is the batch mean, $v_b^2$ is the batch variance, $\gamma$ is the scaling parameter, and $\beta$ is the biasing parameter.

## 3.2 REVISITING EXISTING RBN

Inspired by the remarkable success of normalization techniques in traditional DNNs, endeavors have been made to develop Riemannian normalization approaches tailored for manifolds. Here we recap some representative methods. However, we note that none of the existing methods effectively handle the first and second moments in a principled manner.

Brooks et al. (2019b) introduced RBN over SPD manifolds under AIM, with operations defined as:

$$\text{Centering from geometric mean } M : \forall i \leq N, \bar{P}_i \leftarrow M^{-\frac{1}{2}} P_i M^{-\frac{1}{2}}, \tag{7}$$

$$\text{Biasing towards SPD parameter } B : \forall i \leq N, \hat{P}_i \leftarrow B^{\frac{1}{2}} \bar{P}_i B^{\frac{1}{2}}, \tag{8}$$

where $\{P_{i...N}\}$ are SPD matrices, and $M$ are their Fréchet mean under AIM. Define a map as

$$\Gamma_{P \to Q}(S) = \text{Exp}_Q \left[ \text{PT}_{P \to Q} \left( \text{Log}_P(S) \right) \right], \tag{9}$$

where $P, Q, S \in \mathcal{S}_{++}^n$, and $\text{Exp}, \text{Log}, \text{PT}$ are Riemannian exponential map, Riemannian logarithmic map, and parallel transportation along the geodesic, respectively. Then under AIM, Eqs. (7) and (8) can be more generally expressed as

$$\Gamma_{I \to B}[\Gamma_{M \to I}(P_i)]. \tag{10}$$

However, Eqs. (7) and (8) only consider the Riemannian mean[2] and does not consider the Riemannian variance. To remedy this limitation, Kobler et al. (2022b) further improved the RBN to enable control over the Riemannian variance. The key operation is formulated as

$$\forall i \leq N, \bar{P}_i \leftarrow \Gamma_{I \to B}[(\Gamma_{M \to I}(P_i))^{\frac{s}{v}}], \tag{11}$$

where $v^2$ is the Fréchet variance, $s \in \mathbb{R}$ is a scaling factor. However, this method is still limited to SPD manifolds under AIM. In parallel, Chakraborty (2020, Algs. 1-2) proposed a general framework for Riemannian homogeneous spaces based on Eq. (10) by considering both first and second moments. However, as stated in Chakraborty (2020, Sec. 3.1), Eq. (10) does not generally guarantee the control over Riemannian mean, resulting in agnostic Riemannian statistics. To mitigate this limitation, Chakraborty (2020, Algs. 3-4) further proposed normalization over the matrix Lie group. However, the discussion is limited to a certain distance, limiting the applicability of their method. On the other hand, Lou et al. (2020) proposed an RBN based on a variance of Eq. (10). Although their framework encompasses the standard Euclidean BN when the latent geometry is the standard Euclidean geometry, their approach suffers from the same problem of agnostic Riemannian statistics on general manifolds.

In summary, prevailing Riemannian normalization approaches lack a principled guarantee for controlling the first and second-order statistics. In contrast, as will be elucidated, our method can govern first and second-order statistics for all Lie groups. We summarize the above RBN methods in Tab. 2.

## 4 RIEMANNIAN NORMALIZATION ON LIE GROUPS

In this section, the neutral element in the Lie group $\mathcal{M}$ is denoted as $E$. Notably, the neutral element $E$ is not necessarily the identity matrix. We first clarify the essential properties of Euclidean BN and then present our normalization method, tailored for Lie groups.

---

[2]Although not mentioned in the original paper, Eqs. (7) and (8) can guarantee the mean of the resulting samples under AIM, as Eqs. (7) and (8) are actions of $\text{GL}(n)$.

Table 2: Summary of some representative RBN methods.

| Methods | Involved Statistics | Controllable Mean | Controllable Variance | Application Scenarios |
|---|---|---|---|---|
| SPDBN (Brooks et al., 2019b) | Mean | ✓ | N/A | SPD manifolds under AIM |
| SPDBN (Kobler et al., 2022b) | Mean+Variance | ✓ | ✓ | SPD manifolds under AIM |
| Chakraborty (2020, Algs. 1-2) | Mean+Variance | ✗ | ✗ | Riemannian homogeneous space |
| Chakraborty (2020, Algs. 3-4) | Mean+Variance | ✓ | ✓ | A certain Lie group structure and distance |
| RBN (Lou et al., 2020, Alg. 2) | Mean+Variance | ✗ | ✗ | Geodesically complete manifolds |
| Ours | Mean+Variance | ✓ | ✓ | General Lie groups |

Two key points regarding Euclidean BN, as expressed by Eq. (6), are worth highlighting: (a) The standard BN (Ioffe & Szegedy, 2015) implicitly assumes a Gaussian distribution and can effectively normalize and transform the latent Gaussian distribution; (b) The centering and biasing operations control the mean, while the scaling controls the variance. Therefore, extending BN into Lie groups requires defining Gaussian distribution, centering, biasing, and scaling on Lie groups.

On manifolds, several notions of Gaussian distribution have been proposed (Pennec, 2004; Chakraborty & Vemuri, 2019; Barbaresco, 2021). In this work, we adopt the definition from Chakraborty & Vemuri (2019), which characterizes a Gaussian distribution on the Lie group $\mathcal{M}$ with a mean parameter $M \in \mathcal{M}$ and variance $\sigma^2$. This distribution is denoted as $\mathcal{N}(M, \sigma^2)$, and its Probability Density Function (P.D.F.) is defined as[3]:

$$p\left(X \mid M, \sigma^2\right) = k(\sigma) \exp\left(-\frac{\mathrm{d}(X, M)^2}{2\sigma^2}\right),$$ 
(12)

where $k(\sigma)$ is the normalizing constant, $\exp(\cdot)$ is the scalar exponentiation, and $\mathrm{d}(\cdot, \cdot)$ is the geodesic distance. On Lie groups, the natural counterparts of addition and subtraction in Eq. (6) are group operations. Therefore, centering and biasing on Lie groups can be defined by Lie group left translation. Additionally, scaling can be defined by scaling on the tangent space at the Riemannian mean.

Now, we can extend Eq. (6) into Lie groups $\mathcal{M}$. For a batch of activation $\{P_{i...N} \in \mathcal{M}\}$, we define the key operations of Lie group BN (LieBN) as:

$$\text{Centering to the neutral element } E: \forall i \leq N, \bar{P}_i \leftarrow L_{M_{\odot}^{-1}}(P_i),$$ 
(13)

$$\text{Scaling the dispersion: } \forall i \leq N, \hat{P}_i \leftarrow \mathrm{Exp}_E\left[\frac{s}{\sqrt{v^2 + \epsilon}} \mathrm{Log}_E(\bar{P}_i)\right],$$ 
(14)

$$\text{Biasing towards parameter } B \in \mathcal{M}: \forall i \leq N, \tilde{P}_i \leftarrow L_B(\hat{P}_i),$$ 
(15)

where $M$ is the Fréchet mean, $v^2$ is the Fréchet variance, $M_{\odot}^{-1} \in \mathcal{M}$ is the group inverse of $M$, $L_{M_{\odot}^{-1}}$ and $L_G$ are left translations ($L_G(P_i) = G \odot P_i$), and $s \in \mathbb{R}/\{0\}$ is a scaling parameter. To clarify the effect of our method in controlling mean and variance, we first present the following two propositions, one related to population statistics and the other related to sample statistics.

**Proposition 4.1** (Population). [↓] *Given a random point $X$ over $\mathcal{M}$, and the Gaussian distribution $\mathcal{N}(M, v^2)$ defined in Eq. (12), we have the following properties for the population statistics:*

1. *(MLE of $M$) Given $\{P_{i...N} \in \mathcal{M}\}$ i.i.d. sampled from $\mathcal{N}(M, v^2)$, the maximum likelihood estimator (MLE) of $M$ is the sample Fréchet mean.*

2. *(Homogeneity) Given $X \sim \mathcal{N}(M, v^2)$ and $B \in \mathcal{M}$, $L_B(X) \sim \mathcal{N}(L_B(M), v^2)$*

**Proposition 4.2** (Sample). [↓] *Given $N$ samples $\{P_{i...N} \in \mathcal{M}\}$, denoting $\phi_s(P_i) = \mathrm{Exp}_E\left[s \mathrm{Log}_E(P_i)\right]$, we have the following properties for the sample statistics:*

$$\text{Homogeneity of the sample mean: } \mathrm{FM}\{L_B(P_i)\} = L_B(\mathrm{FM}\{P_i\}), \forall B \in \mathcal{M},$$ 
(16)

$$\text{Controllable dispersion from } E: \sum_{i=1}^{N} w_i \, \mathrm{d}^2(\phi_s(P_i), E) = s^2 \sum_{i=1}^{N} w_i \, \mathrm{d}^2(P_i, E),$$ 
(17)

*where $\{w_{1...N}\}$ are weights satisfying a convexity constraint, i.e.,$\forall i, w_i > 0$ and $\sum_i w_i = 1$.*

---

[3]When $\mathcal{M}$ corresponds to $\mathbb{R}$ equipped with the standard Euclidean metric, Eq. (12) reduces to the Euclidean Gaussian distribution.

Prop. 4.1 and Eq. (16) imply that our centering and biasing in Eqs. (13) and (15) can control the sample and population mean. As the post-centering mean is $E$, Eq. (17) implies that Eq. (14) can control the dispersion. Although the population variance after Eq. (14) is generally agnostic, in some cases such as SPD manifolds under LEM and LCM, Eq. (14) can normalize the population variance. Please refer to App. C for technical details.

Similar to Ioffe & Szegedy (2015), we use moving averages to update running statistics. Now, we present our general framework of LieBN in Alg. 1. Importantly, when $\mathcal{M} = \mathbb{R}^n$, Alg. 1 is reduced to the standard Euclidean BN. More details are exposed in App. D.

*Remark* 4.3. The MLE of the mean of the Gaussian distribution in Eq. (12) have been examined in several previous works (Said et al., 2017; Chakraborty & Vemuri, 2019; Chakraborty, 2020). However, these studies primarily focus on particular manifolds or specific metrics. ***In contrast, our contribution lies in presenting a universally applicable result for all Lie groups with left-invariant metrics.*** While Eq. (12) briefly appeared in Kobler et al. (2022b), the authors only focus on SPD manifolds under AIM. The transformation of the population under their proposed RBN remains unexplored as well. Besides, while Chakraborty (2020) analyzed the population properties for their RBN over matrix Lie groups, their results were confined within a specific distance. In contrast, our work provides a more extensive examination, encompassing both population and sample properties of our LieBN in a general manner. Besides, all the discussion about our LieBN can be readily transferred to right-invariant metrics. This paper focuses on LieBN based on left-invariant metrics.

---

**Algorithm 1:** Lie Group Batch Normalization (LieBN) Algorithm

---

**Input** : A batch of activations $\{P_{1\ldots N} \in \mathcal{M}\}$, a small positive constant $\epsilon$, and
momentum $\gamma \in [0, 1]$
running mean $M_r = E$, running variance $v_r^2 = 1$,
biasing parameter $B \in \mathcal{M}$, scaling parameter $s \in \mathbb{R}/\{0\}$,

**Output** : Normalized activations $\{\tilde{P}_{1\ldots N}\}$

---

**if** *training* **then**
  Compute batch mean $M_b$ and variance $v_b^2$ of $\{P_{1\ldots N}\}$;
  Update running statistics $M_r \leftarrow \mathrm{WFM}(\{1 - \gamma, \gamma\}, \{M_r, M_b\}), v_r^2 \leftarrow (1 - \gamma)v_r^2 + \gamma v_b^2$;
**end**
**if** *training* **then** $M \leftarrow M_b, v^2 \leftarrow v_b^2$;
**else** $M \leftarrow M_r, v^2 \leftarrow v_r^2$;
**for** $i \leftarrow 1$ **to** $N$ **do**
  Centering to the neutral element $E$: $\bar{P}_i \leftarrow L_{M_\odot^{-1}}(P_i)$
  Scaling the dispersion: $\hat{P}_i \leftarrow \mathrm{Exp}_E \left[ \frac{s}{\sqrt{v^2+\epsilon}} \mathrm{Log}_E(\bar{P}_i) \right]$
  Biasing towards parameter $B$: $\tilde{P}_i \leftarrow L_B(\hat{P}_i)$
**end**

---

## 5 LieBN on the Lie Groups of SPD Manifolds

This section showcases our Alg. 1 on SPD manifolds. Firstly, we extend the current Lie groups on SPD manifolds by the concept of deformation, resulting in three families of parameterized Lie groups. Subsequently, we construct LieBN layers based on these generalized Lie groups.

### 5.1 Deformed Lie Groups of SPD Manifolds

As shown in Tab. 1, there are three types of Lie groups on SPD manifolds, each equipped with a left-invariant metric. These metrics include $(\alpha, \beta)$-AIM, $(\alpha, \beta)$-LEM, and LCM. For clarity, we denote the group operations w.r.t. $(\alpha, \beta)$-AIM, $(\alpha, \beta)$-LEM and LCM as $\odot^{\mathrm{AI}}$, $\odot^{\mathrm{LE}}$ and $\odot^{\mathrm{LC}}$, respectively.

In Thanwerdas & Pennec (2019b), $(\alpha, \beta)$-AIM is further extended into three-parameters families of metrics by the pullback of matrix power function $\mathrm{P}_\theta(\cdot)$ and scaled by $\frac{1}{\theta^2}$, denoted as $(\theta, \alpha, \beta)$-AIM. The power function serves as a deformation wherein $(\theta, \alpha, \beta)$-AIM encompasses $(\alpha, \beta)$-AIM when $\theta = 1$, and includes $(\alpha, \beta)$-LEM as $\theta$ approaches 0 (Thanwerdas & Pennec, 2019a). Inspired by the deforming utility of the power function, we define the power-deformed metrics of $(\alpha, \beta)$-LEM and LCM as the pullback metrics by $\mathrm{P}_\theta$ and scaled by $\frac{1}{\theta^2}$. We denote these two metrics as $(\theta, \alpha, \beta)$-LEM and $\theta$-LCM, respectively. We have the following results w.r.t. the deformation.

**Proposition 5.1** (Deformation). [↓] $(\theta, \alpha, \beta)$-*LEM is equal to* $(\alpha, \beta)$-*LEM.* $\theta$-*LCM interpolates between* $\tilde{g}$-*LEM* $(\theta = 0)$ *and LCM* $(\theta = 1)$, *with* $\tilde{g}$-*LEM defined as*

$$\langle V, W \rangle_P = \tilde{g}(\mathrm{mlog}_{*,P}(V), \mathrm{mlog}_{*,P}(W)), \forall P \in \mathcal{S}_{++}^n, \forall V, W \in T_P \mathcal{S}_{++}^n, \qquad (18)$$

*where* $\tilde{g}(V_1, V_2) = \frac{1}{2}\langle V_1, V_2 \rangle - \frac{1}{4}\langle \mathbb{D}(V_1), \mathbb{D}(V_2)\rangle$, $\mathbb{D}(V_i)$ *is a diagonal matrix consisting of the diagonal elements of* $V_i$, *and* $\mathrm{mlog}_{*,P}$ *is the differential map at* $P$.

As $(\theta, \alpha, \beta)$-LEM is equal to $(\alpha, \beta)$-LEM, in the following, we focus on $(\alpha, \beta)$-LEM, $(\theta, \alpha, \beta)$-AIM, and $\theta$-LCM. As a diffeomorphism, $P_\theta$ also can pull back the group operation $\odot^{\mathrm{AI}}$ and $\odot^{\mathrm{LC}}$, denoted as $\odot^{\theta\text{-AI}}$ and $\odot^{\theta\text{-LC}}$. We have the following proposition on the invariance.

**Proposition 5.2** (Invariance). [↓] $(\theta, \alpha, \beta)$-*AIM is left-invariant w.r.t.* $\odot^{\theta\text{-AI}}$, *while* $\theta$-*LCM is bi-invariant w.r.t.* $\odot^{\theta\text{-LC}}$.

### 5.2 LieBN on SPD Manifolds

Now, we showcase our LieBN framework illustrated in Alg. 1 on SPD manifolds. As discussed in Sec. 5.1, there are three families of left-invariant metrics, namely $(\theta, \alpha, \beta)$-AIM, $(\alpha, \beta)$-LEM, and $\theta$-LCM. Since all three metric families are pullback metrics, the LieBN based on these metrics can be simplified and calculated in the co-domain. We denote Alg. 1 as

$$\mathrm{LieBN}(P_i; B, s, \epsilon, \gamma), \forall p_i \in \{P_{1...N} \in \mathcal{M}\}. \qquad (19)$$

Then we can obtain the following theorem.

**Theorem 5.3.** [↓] *Given a Lie group* $\mathcal{M}_1$, *a Lie group* $\mathcal{M}_2$ *with a left-invariant metric* $g^2$, *and a diffeomorphism* $f : \mathcal{M}_1 \to \mathcal{M}_2$, *then* $f$ *induces a left-invariant metric* $g^1$ *on* $\mathcal{M}_1$, *denoted as* $g^1 = f^* g^2$. *For a batch of activation* $\{P_{1...N}\}$ *in* $\mathcal{M}_1$, $\mathrm{LieBN}(P_i; B, s, \epsilon, \gamma)$ *in* $\mathcal{M}_1$ *can be calculated in* $\mathcal{M}_2$ *by the following process:*

$$\text{Mapping data into } \mathcal{M}_2 : \bar{P}_i = f(P_i), \bar{B} = f(B), \qquad (20)$$

$$\text{Calculating LieBN in } \{\mathcal{M}_2, g^2\} : \hat{P}_i = \mathrm{LieBN}(\bar{P}_i; \bar{B}, s, \epsilon, \gamma), \qquad (21)$$

$$\text{Mapping the normalized data back to } \mathcal{M}_1 : \tilde{P}_i = f^{-1}(\hat{P}_i), \qquad (22)$$

Given a metric $g$ on $\mathcal{S}_{++}^n$, the power-deformed metric $\tilde{g} = \frac{1}{\theta^2} \mathrm{P}_\theta^* g$ is equal to $\mathrm{P}_\theta^*(\frac{1}{\theta^2} g)$. Therefore, the LieBN under $\tilde{g}$ can be calculated by the LieBN under $\frac{1}{\theta^2} g$. Besides, as the Christoffel symbols remain the same under constant scaling, the LieBNs under $\frac{1}{\theta^2} g$ and $g$ only differ in the variance. We denote $g^{(\alpha,\beta)\text{-AI}}$ and $g^{(\theta,\alpha,\beta)\text{-AI}}$ as the metric tensors of $(\alpha, \beta)$-AIM and $(\theta, \alpha, \beta)$-AIM, respectively Based on the above discussions, the computations of the LieBN under $g^{(\theta,\alpha,\beta)\text{-AI}}$ are reduced to the LieBN under $\frac{1}{\theta^2} g^{(\alpha,\beta)\text{-AI}}$. Furthermore, as shown in Chen et al. (2023c), $(\alpha, \beta)$-LEM is a pullback metric from the Euclidean space $\mathcal{S}^n$ of symmetric matrices, while the proof of Prop. 5.2 indicates that $\theta$-LCM is a pullback metric from the Euclidean space $\mathcal{L}^n$ of lower triangular matrices. Note that in the Euclidean space $\mathcal{S}^n$ or $\mathcal{L}^n$, as shown in App. D, Eq. (21) simplifies to the standard Euclidean BN. We denote $P, Q, P_1$ and $P_2$ as points in the codomain ($\mathcal{S}_{++}^n$ with $(\alpha, \beta)$-AIM for $(\theta, \alpha, \beta)$-AIM, $\mathcal{S}^n$ for $(\alpha, \beta)$-LEM, and $\mathcal{L}^n$ for $\theta$-LCM, respectively). We summarize all the necessary ingredients in Tab. 3 for calculating LieBN on SPD manifolds. Note that for $(\theta, \alpha, \beta)$-AIM, our scaling operation defined in Eq. (14) encompasses the scaling operation in Kobler et al. (2022b, Eq. (9)) as a special case, when $(\theta, \alpha, \beta) = (1, 1, 0)$.

## 6 Experiments

In this section, we implement our three families of LieBN to SPD neural networks. Following the previous work (Huang & Van Gool, 2017; Brooks et al., 2019b; Kobler et al., 2022a), we adopt three different applications: radar recognition on the Radar dataset (Brooks et al., 2019b), human action recognition on the HDM05 (Müller et al., 2007) and FPHA (Garcia-Hernando et al., 2018) datasets, and EEG classification on the Hinss2021 dataset (Hinss et al., 2021). More details on datasets and hyper-parameters are exposed in App. G. Besides SPD neural networks, we also implement LieBN on special orthogonal groups and present some preliminary experiments (see App. H).

**Implementation details:** Note that our LieBN layers are architecture-agnostic and can be applied to any existing SPD neural network. In this paper, we focus on two network architectures: SPDNet

Table 3: Key operators in calculating LieBN on SPD manifolds.

| Metric | $(\theta,\alpha,\beta)$-AIM | $(\alpha,\beta)$-LEM | $\theta$-LCM |
|---|---|---|---|
| Pullback Map | $P_\theta$ | mlog | $P_\theta \circ \psi_{\mathrm{LC}}$ |
| Codomain | $\{\mathcal{S}^n_{++}, \odot^{\mathrm{AI}}, \frac{1}{\theta^2}g^{(\alpha,\beta)\text{-}\mathrm{AI}}\}$ | $\{\mathcal{S}^n, \langle\cdot,\cdot\rangle^{(\alpha,\beta)}\}$ | $\{\mathcal{L}^n, \frac{1}{\theta^2}\langle\cdot,\cdot\rangle\}$ |

| | | $(\theta,\alpha,\beta)$-AIM | $(\alpha,\beta)$-LEM | $\theta$-LCM |
|---|---|---|---|---|
| Riemannian and Lie group operators in the codomain | $L_Q(P)$ | $KPK^\top$ | $P+Q$ | $P+Q$ |
| | $L_{Q_\odot^{-1}}(P)$ | $K^{-1}PK^{-\top}$ | $P-Q$ | $P-Q$ |
| | $\mathrm{Exp}_E\left[s\,\mathrm{Log}_E(P)\right]$ | $P^s$ | $sP$ | $sP$ |
| | FM | Karcher Flow | Arithmetic average | Arithmetic average |
| | $\mathrm{WFM}(\{\gamma,1-\gamma\},\{P_1,P_2\})$ | $P_2^{\frac{1}{2}}\left(P_2^{-\frac{1}{2}}P_1P_2^{-\frac{1}{2}}\right)^\gamma P_2^{\frac{1}{2}}$ | Arithmetic weighted average | Arithmetic weighted average |

Table 4: 10-fold average results of SPDNet with and without SPDBN or LieBN on the Radar, HDM05, and FPHA datasets. For simplicity, LieBN-Metric-($\theta$) is abbreviated as Metric-($\theta$). For the LieBN under each metric, if the LieBN induced by the standard metric ($\theta = 1$) is not saturated, we report the LieBN under the deformed metric in the rightmost columns of the table.

(a) Radar dataset.

| Method | SPDNet | SPDNetBN | AIM-(1) | LEM-(1) | LCM-(1) | LCM-(-0.5) |
|---|---|---|---|---|---|---|
| Fit Time (s) | 0.98 | 1.56 | 1.62 | 1.28 | 1.11 | 1.43 |
| Mean±STD | 93.25±1.10 | 94.85±0.99 | **95.47±0.90** | 94.89±1.04 | 93.52±1.07 | 94.80±0.71 |
| Max | 94.4 | 96.13 | 96.27 | **96.8** | 95.2 | 95.73 |

(b) HDM05 and FPHA datasets.

| | Method | SPDNet | SPDNetBN | AIM-(1) | LEM-(1) | LCM-(1) | AIM-(1.5) | LCM-(0.5) |
|---|---|---|---|---|---|---|---|---|
| HDM05 | Fit Time (s) | 0.57 | 0.97 | 1.14 | 0.87 | 0.66 | 1.46 | 1.01 |
| | Mean±STD | 59.13±0.67 | 66.72±0.52 | 67.79±0.65 | 65.05±0.63 | 66.68±0.71 | 68.16±0.68 | **70.84±0.92** |
| | Max | 60.34 | 67.66 | 68.75 | 66.05 | 68.52 | 69.25 | **72.27** |
| FPHA | Fit Time (s) | 0.32 | 0.62 | 0.80 | 0.55 | 0.39 | 1.03 | 0.65 |
| | Mean±STD | 85.59±0.72 | 89.33±0.49 | 89.70±0.51 | 86.56±0.79 | 77.64±1.00 | **90.39±0.66** | 86.33±0.43 |
| | Max | 86 | 90.17 | 90.5 | 87.83 | 79 | **92.17** | 87 |

(Huang & Van Gool, 2017) for the Radar, HDM05, and FPHA datasets; and TSMNet (Kobler et al., 2022a) for the Hinss2021 dataset. For the SPDNet architecture, we compare our LieBN with SPDNetBN (Brooks et al., 2019b), which applies the SPDBN (Eqs. (7) and (8)) to SPDNet. Consistent with SPDNetBN, we apply our LieBN after each transformation layer (BiMap layer in App. B). In the EEG application, the state-of-the-art Riemannian method is TSMNet with SPD domain-specific momentum batch normalization (TSMNet+SPDDSMBN) (Kobler et al., 2022a), which is a domain adaptation version of Kobler et al. (2022b). For a fair comparison, we also implement a domain-specific momentum LieBN, referred to as DSMLieBN (detailed in App. E). Following Kobler et al. (2022a), we apply our DSMLieBN before the LogEig layer (detailed in Appendix B) in TSMNet. We use the standard cross-entropy loss as the training objective and optimize the parameters with the Riemannian AMSGrad optimizer (Bécigneul & Ganea, 2018). The network architectures are represented as $\{d_0, d_1, \ldots, d_L\}$, where the dimension of the parameter in the $i$-th BiMap layer is $d_i \times d_{i-1}$. The experiments are conducted with a learning rate of $5e^{-3}$, batch size of 30, and training epoch of 200 on the Radar, HDM05, and FPHA datasets. For the Hinss2021 dataset, following Kobler et al. (2022a), we use a learning rate of $1e^{-3}$ with a weight decay of $1e^{-4}$, a batch size of 50, and a training epoch of 50. All experiments use an Intel Core i9-7960X CPU with 32GB RAM and an NVIDIA GeForce RTX 2080 Ti GPU. Evaluation methods are explained in App. G.3.

## 6.1 EXPERIMENTAL RESULTS

For each family of LieBN or DSMLieBN, we report two representatives: the standard one induced from the standard metric ($\theta = 1$), and the one induced from the deformed metric with selected $\theta$. *If the standard one is already saturated, we only report the results of the standard ones.*

**Application to SPDNet:** As SPDNet is the most classic SPD network, we apply our LieBN to SPDNet on the Radar, HDM05, and FPHA datasets. Additionally, we compare our method with SPDNetBN, which applies the SPDBN in Eqs. (7) and (8) to SPDNet. Following Brooks et al.

Table 5: Cross-validation results of TSMNet with SPDDSMBN and DSMLieBN on the Hinss dataset. For simplicity, DSMLieBN-Metric-($\theta$) is abbreviated as Metric-($\theta$). For the DSMLieBN under each metric, if the DSMLieBN induced by the standard metric ($\theta = 1$) is not saturated, we report the DSMLieBN under the deformed metric at the bottom rows of the table.

(a) Inter-session classification

| Method | Fit Time (s) | Mean±STD |
|---|---|---|
| SPDDSMBN | 0.16 | 54.12±9.87 |
| AIM-(1) | 0.16 | **55.10±7.61** |
| LEM-(1) | 0.13 | 54.95±10.09 |
| LCM-(1) | 0.10 | 51.54±6.88 |
| LCM-(0.5) | 0.15 | 53.11±5.65 |

(b) Inter-subject classification

| Method | Fit Time (s) | Mean±STD |
|---|---|---|
| SPDDSMBN | 7.74 | 50.10±8.08 |
| AIM-(1) | 6.94 | 50.04±8.01 |
| LEM-(1) | 4.71 | 50.95±6.40 |
| LCM-(1) | 3.59 | 51.86±4.53 |
| AIM-(-0.5) | 8.71 | **53.97±8.78** |

(2019b); Chen et al. (2023d), we use the architectures of $\{20, 16, 8\}$, $\{93, 30\}$, and $\{63, 33\}$ for the Radar, HDM05 and FPHA datasets, respectively. The 10-fold average results, including the average training time (s/epcoh), are summarized in Tab. 4. We have three key observations regarding the choice of metrics, deformation, and training efficiency. **The choice of metrics:** The metric that yields the most effective LieBN layer differs for each dataset. Specifically, the optimal LieBN layers on these three datasets are the ones induced by AIM-(1), LCM-(0.5), and AIM-(1.5), respectively, **which improves the performance of SPDNet by 2.22%, 11.71%, and 4.8%**. Additionally, although the LCM-based LieBN performs worse than other LieBN variants on the Radar and FPHA datasets, it exhibits the best performance on the HDM05 dataset. These observations highlight the advantage of the generality of our LieBN approach. **The effect of deformation:** Deformation patterns also vary across datasets. Firstly, the standard AIM is already saturated on the Radar dataset. Secondly, as indicated in Tab. 4, an appropriate deformation factor $\theta$ can further enhance the performance of LieBN. Notably, even though the LieBN induced by LCM-(1) impedes the learning of SPDNet on the FPHA datasets, it can improve performance when an appropriate deformation factor $\theta$ is applied. These findings highlight the utility of the deforming geometry of the SPD manifold. **Efficiency:** Our LieBN achieves comparable or even better efficiency than SPDNetBN, although compared with SPDNetBN, our LieBN places additional consideration on variance. Particularly, the LieBN induced by standard LEM or LCM exhibits better efficiency than SPDNetBN. Even with deformation, the LCM-based LieBN is still comparable with SPDNetBN in terms of efficiency. This phenomenon could be attributed to the fast and simple computation of LCM and LEM.

**Application to EEG classification:** We evaluate our method on the architecture of TSMNet for two tasks, inter-session and inter-subject EEG classification. Following Kobler et al. (2022a), we adopt the architecture of $\{40, 20\}$. Compared to the SPDDSMBN, TSMNet+DSMLieBN-AIM obtains the highest average scores of 55.10% and 53.97% in inter-session and -subject transfer learning, improving the SPDDSMBN by 0.98% and 3.87%, respectively. In the inter-subject scenario, the advantage of the efficiency of our LieBN over SPDDSMBN is more obvious. Specifically, both the LEM- and LCM-based DSMLieBN achieve similar or better performance compared to SPDDSMBN, while requiring considerably less training time. For example, DSMLieBN-LCM-(1) achieves better results with only half the training time of SPDDSMBN on inter-subject tasks. Interestingly, under the standard AIM, the sole difference between SPDDSMBN and our DSMLieBN is the way of centering and biasing. SPDDSMBN applies the inverse square root and square root to fulfill centering and biasing, while AIM-induced LieBN uses more efficient Cholesky decomposition. As such, the DSMLieBN induced by the standard AIM is more efficient than SPDDSMBN, particularly on the inter-subject task.

## 7 CONCLUSIONS

This paper proposes a novel framework called LieBN, enabling batch normalization over Lie groups. Our LieBN can effectively normalize both the sample and population statistics. Besides, we generalize the existing Lie groups on SPD manifolds and showcase our framework on the parameterized Lie groups of SPD manifolds. Extensive experiments demonstrate the advantage of our LieBN.

There are several other types of Lie groups in machine learning, such as special Euclidean groups. As a future avenue, we shall extend our LieBN to other Lie groups.

ACKNOWLEDGMENTS

This work was partly supported by the MUR PNRR project FAIR (PE00000013) funded by the NextGenerationEU, by the EU Horizon project ELIAS (No. 101120237), and by a gift donation from Cisco. The authors also gratefully acknowledge financial support from the China Scholarship Council (CSC).

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

APPENDIX CONTENTS

## A    NOTATIONS

For better clarity, we summarize all the notations used in this paper in Tab. 6.

Table 6: Summary of notations.

| Notation | Explanation |
| --- | --- |
| $\{\mathcal{M}, \odot, g\}$ or abbreviated as $\mathcal{M}$ | A Lie group with a group operation $\odot$ and left-invariant Riemannian metric $g$ |
| $P_\odot^{-1}$ | Group inverse of $P \in \mathcal{M}$ |
| $T_P \mathcal{M}$ | The tangent space at $P \in \mathcal{M}$ |
| $g_p(\cdot, \cdot)$ or $\langle \cdot, \cdot \rangle_P$ | The Riemannian metric at $P \in \mathcal{M}$ |
| $\| \cdot \|_P$ | The norm induced by $\langle \cdot, \cdot \rangle_P$ on $T_P \mathcal{M}$ |
| $\mathrm{d}(\cdot, \cdot)$ | Geodesic distance |
| FM | Fréchet mean |
| WFM | Weighted Fréchet mean |
| $\mathrm{Log}_P$ | The Riemannian logarithm at $P$ |
| $\mathrm{PT}_{P \to Q}$ | The Riemannian parallel transportation along the geodesic connecting $P$ and $Q$ |
| $L_P$ | The Lie group left translation by $P \in \mathcal{M}$ |
| $f_{*,P}$ | The differential map of the smooth map $f$ at $P \in \mathcal{M}$ |
| $f^* g$ | The pullback metric by $f$ from $g$ |
| $\mathcal{S}_{++}^n$ | The SPD manifold |
| $\mathcal{S}^n$ | The Euclidean space of symmetric matrices |
| $\mathcal{L}^n$ | The Euclidean space of lower triangular matrices |
| $\langle \cdot, \cdot \rangle$ | The standard Frobenius inner product |
| $\| \cdot \|_F$ | The norm induced by the standard Frobenius inner product |
| **ST** | $\mathbf{ST} = \{(\alpha, \beta) \in \mathbb{R}^2 \mid \min(\alpha, \alpha + n\beta) > 0\}$ |
| $\langle \cdot, \cdot \rangle^{(\alpha, \beta)}$ | The $O(n)$-invariant Euclidean inner product |
| $\| \cdot \|^{(\alpha, \beta)}$ | The norm induced by $O(n)$-invariant Euclidean inner product |
| $g^{(\alpha, \beta)\text{-AI}}$ | The Riemannian metric tensor of $(\alpha, \beta)$-AIM |
| $g^{(\theta, \alpha, \beta)\text{-AI}}$ | The Riemannian metric tensor of $(\theta, \alpha, \beta)$-AIM |
| $\odot^{\mathrm{AI}}$ | The group operation w.r.t. AIM |
| $\odot^{\mathrm{LE}}$ | The group operation w.r.t. LEM |
| $\odot^{\mathrm{LC}}$ | The group operation w.r.t. LCM |
| $\mathcal{N}(M, \sigma^2)$ | Riemannian Gaussian distribution |
| exp | Scalar exponentiation |
| mlog | Matrix logarithm |
| mexp | Matrix exponentiation |
| Chol | Cholesky decomposition |
| Dlog | The diagonal element-wise logarithm |
| $\psi_{\mathrm{LC}}$ | $\mathrm{Dlog} \circ \mathrm{Chol}$ |
| $\lfloor \cdot \rfloor$ | The strictly lower triangular part of a square matrix |
| $\mathbb{D}(\cdot)$ | A diagonal matrix with diagonal elements from a square matrix |
| $\mathrm{P}_\theta(\cdot)$ or $(\cdot)^\theta$ | Matrix power function |

## B    BASIC LAYERS IN SPDNET AND TSMNET

SPDNet (Huang & Van Gool, 2017) is the most classic SPD neural network. SPDNet mimics the conventional densely connected feedforward network, consisting of three basic building blocks

$$\text{BiMap layer: } S^k = W^k S^{k-1} W^{k\top}, \text{ with } W^k \text{ semi-orthogonal}, \tag{23}$$

$$\text{ReEig layer: } S^k = U^{k-1} \max(\Sigma^{k-1}, \epsilon I_n) U^{k-1\top}, \text{ with } S^{k-1} = U^{k-1} \Sigma^{k-1} U^{k-1\top}, \tag{24}$$

$$\text{LogEig layer: } S^k = \log(S^{k-1}). \tag{25}$$

where $\max(\cdot)$ is element-wise maximization. BiMap and ReEig mimic transformation and non-linear activation, while LogEig maps SPD matrices into the tangent space at the identity matrix for classification.

TSMNet (Kobler et al., 2022a) can be illustrate as $f_{tc} \to f_{sc} \to f_{BiMap} \to f_{ReEig} \to f_{LogEig}$, where $f_{tc}$ and $f_{sc}$ denote temporal and spatial convolution, respectively.

## C    STATISTICAL RESULTS OF SCALING IN THE LIEBN

In this section, we will show the effect of our scaling (Eq. (14)) on the population. We will see that while the resulting population variance is generally agnostic, it becomes analytic under certain

circumstances, such as SPD manifolds under LEM or LCM. As a result, Eq. (14) can normalize and transform the latent Gaussian distribution.

To simplify, let $\phi_s(P) = \text{Exp}_E[s\text{Log}_E(P)]$. Similar to the main paper, $\mathcal{M}$ denotes a Lie group with a left-invariant metric. First, we present a lemma on the resulting P.D.F. of a random point transformed by $\phi_s$.

**Lemma C.1.** *Given a random point $X$ distributed over $\mathcal{M}$ with P.D.F. $p_X$, the P.D.F of $Y = \phi_s(X)$ is given by:*

$$p_Y(Q) = \Delta p_X(\phi_s^{(-1)}(Q)). \tag{26}$$

*where $\Delta = \dfrac{|\phi_{s*}^{-1}|}{\left|L_{\phi_s^{-1}(Q)\odot Q^{-1}*}\right|}$. Here $|\cdot|$ denotes the determinant, and $\phi_{s*}^{-1}$ and $L_{\phi_s^{-1}(Q)\odot Q^{-1}*}$ are the differentials.*

*Proof.* For the sake of simplicity, we will denote $\phi_s$ as $\phi$ throughout this proof. The volume element w.r.t. a left-invariant metric is the Haar measure (Pennec & Ayache, 1998, Sec. 3.2):

$$\text{d}_L\,\mathcal{M}(P) = \frac{dP}{|L_{P*,E}|}, \tag{27}$$

where $|L_{P*,E}|$ is the determinant[4] of the differential of $L_P$ at the neutral element $E$. Then we have

$$
\begin{aligned}
\text{d}_L\,\mathcal{M}(\phi^{-1}(Q)) &= \frac{\text{d}\,\phi^{-1}(Q)}{|L_{\phi^{-1}(Q)*,E}|} \\
&= |(L_{\phi^{-1}(Q)\odot Q^{-1}} \circ L_Q)_{*,E}|^{-1}|\phi_*^{-1}|\,\text{d}\,Q \\
&= \frac{|\phi_*^{-1}|}{|L_{\phi^{-1}(Q)\odot Q^{-1}*}|}\,\text{d}_L\,\mathcal{M}(Q) \\
&= \Delta\,\text{d}_L\,\mathcal{M}(Q).
\end{aligned}
\tag{28}
$$

The probability of $Q = \phi(P)$ in a set $\mathcal{Y} \subset \mathcal{M}$ is

$$
\begin{aligned}
F(\phi(P) \in \mathcal{Y}) &= F(P \in \phi^{-1}(\mathcal{Y})) \\
&= \int_{\phi^{-1}(\mathcal{Y})} p_X(P) \cdot d_L\mathcal{M}(P) \\
&= \int_{\mathcal{Y}} p_X(\phi^{(-1)}(Q))d_L\mathcal{M}(\phi^{(-1)}(Q)) \\
&= \int_{\mathcal{Y}} \Delta p_X(\phi^{(-1)}(Q))d_L\mathcal{M}(Q).
\end{aligned}
\tag{29}
$$

Therefore, the density of $Y = \phi(X)$ is

$$p_Y(Q) = \Delta p_X(\phi^{(-1)}(Q)). \tag{30}$$

$\square$

The above lemma implies that when $\Delta$ is a constant, $Y$ also follows a Gaussian distribution.

**Corollary C.2.** *Following the notations in Lem. C.1, if $\Delta = c$ is a constant and $X \sim \mathcal{N}(E, \sigma^2)$, then $Y$ also follows a Gaussian distribution, i.e.,$Y \sim \mathcal{N}(E, s^2\sigma^2)$*

---

[4]This should be more precisely understood as the determinant of the matrix representation of $L_{P*,E}$ under a local coordinate

*Proof.*

$$
\begin{aligned}
p_Y(Q) &= ck(\sigma) \exp\left(-\frac{\mathrm{d}(\phi_s^{-1}(Q), E)^2}{2\sigma^2}\right) \\
&= k'(\delta) \exp\left(-\frac{\mathrm{d}(\mathrm{Exp}_E \, 1/s \, \mathrm{Log}_E(Q), E)^2}{2\sigma^2}\right) \\
&= k'(\delta) \exp\left(-\frac{\|\mathrm{Log}_E(Q)\|_E^2}{2s^2\sigma^2}\right) \\
&= k'(\delta) \exp\left(-\frac{\mathrm{d}(Q, E)}{2s^2\sigma^2}\right),
\end{aligned}
\tag{31}
$$

where $\|\cdot\|_E$ is the norm of the tangent space at the neutral element $E$. $\qquad\square$

Cor. C.2 implies that when $\Delta = c$, $\phi_s$ can scale the population variance and further transform the Gaussian distribution. Simple computations show that in the standard Euclidean space $\mathbb{R}^n$, $\Delta = 1/s$. Therefore, it is natural to expect that the pullback of $\mathbb{R}^n$ also enjoys constant $\Delta$.

**Proposition C.3.** *Consider an $n$-dimensional Lie group $\mathcal{M}$ pulled back from the standard Euclidean space $\mathbb{R}^n$ by the diffeomorphism $\psi : \mathcal{M} \to \mathbb{R}^n$. In other words, the group operations and Riemannian metric on $\mathcal{M}$ are defined by $\psi$ from $\mathbb{R}^n$. Then $\Delta$ remains constant on $\mathcal{M}$.*

*Proof.* To simplify notation, we denote $\phi_s$ as $\phi$. Under the given assumption, the group addition and Riemannian metric on $\mathcal{M}$ are defined as follows:

$$
\begin{aligned}
\forall P, Q \in \mathcal{M}, P \odot Q &= \psi^{-1}(\psi(P) + \psi(Q)) \\
g &= \psi^* g^{\mathrm{E}},
\end{aligned}
\tag{32}
$$

where $g^{\mathrm{E}}$ is the standard Euclidean metric. Therefore, $\phi$ can be simplified as

$$
\begin{aligned}
\phi(P) &= \mathrm{Exp}_E\left[s \, \mathrm{Log}_E(P)\right] \\
&= \psi^{-1}\left(\tilde{\mathrm{Exp}}_0\left[\psi_{*,E}\left(s\psi_{*,0}^{-1}\tilde{\mathrm{Log}}_0\psi(P)\right)\right]\right) \\
&= \psi^{-1}(s\psi(P)),
\end{aligned}
\tag{33}
$$

where $\tilde{\mathrm{Exp}}$ and $\tilde{\mathrm{Log}}$ are the Riemannian exponential and logarithmic maps in $\mathbb{R}^n$, which are reduced to vector addition and subtraction, respectively. Therefore, the inverse of $\phi$ is

$$
\phi^{-1}(P) = \psi^{-1}\left(1/s\,\psi(P)\right).
\tag{34}
$$

Besides, $L_{\phi^{-1}(Q)\odot Q^{-1}}$ can also be further simplified:

$$
L_{\phi^{-1}(Q)\odot Q^{-1}}(P) = \psi^{-1}\left(1/s\,\psi(Q) - \psi(Q) + \psi(P)\right)
\tag{35}
$$

The differentials of Eqs. (34) and (35) at $Q$ are

$$
\phi_{*,Q}^{-1} = \frac{1}{s}\psi_{*,1/s\psi(Q)}^{-1} \circ \psi_{*,Q},
\tag{36}
$$

$$
L_{\phi^{-1}(Q)\odot Q^{-1}*,Q} = \psi_{*,1/s\psi(Q)}^{-1} \circ \psi_{*,Q}.
\tag{37}
$$

Therefore, $\Delta = 1/s$ for all $Q \in \mathcal{M}$. $\qquad\square$

By Prop. C.3, we can directly obtain the following corollary.

**Corollary C.4.** *Given a Lie group $\mathcal{M}$ pulled back from the Euclidean space, and a random point $X \sim \mathcal{N}(E, \sigma^2)$ over $\mathcal{M}$, $Y = \phi_s(X) \sim \mathcal{N}(E, s^2\sigma^2)$*

In machine learning, several Lie groups are derived by the pullback from the standard Euclidean space. As shown in Chen et al. (2023c) and the proof of Prop. 5.2, $(\alpha, \beta)$-LEM and $\theta$-LCM are pullback metrics from the Euclidean metric. Therefore, for the Lie groups of SPD manifolds w.r.t. $(\alpha, \beta)$-LEM and $\theta$-LCM, Eq. (14) can transform the Gaussian distribution. Specifically, given a random point $X \sim \mathcal{N}(M, \sigma^2)$, Eqs. (13) to (15) transform the Gaussian distribution as:

$$
\mathcal{N}(M, \sigma^2) \to \mathcal{N}(E, \sigma^2) \to \mathcal{N}(E, s^2) \to \mathcal{N}(B, s^2),
\tag{38}
$$

where $M$ and $\sigma$ are employed to normalize $X$, and $\epsilon$ in Eq. (14) is omitted. The above process exactly mirrors the transformation of Gaussian distributions within the framework of standard BN (Ioffe & Szegedy, 2015).

*Remark* C.5. A similar result to our Cor. C.4 was also presented in Chakraborty (2020, Prop. 3). However, in his proof, the author did not account for the Haar measure and only considered the P.D.F., casting doubt on the validity of their results. Additionally, their discussion is limited to matrix Lie groups, specifically under the distance $\mathrm{d}(P,Q) = \|\operatorname{mlog}(P^{-1}Q)\|_{\mathrm{F}}$. In contrast, we rectify their proof and consider general Lie groups.

## D    LieBN as a Natural Generalization of Euclidean BN

The centering and biasing in Euclidean BN correspond to the group action of $\mathbb{R}$. From a geometric perspective, the standard Euclidean metric is invariant under this group operation. Consequently, it is not surprising that our LieBN algorithm formulated in Alg. 1 serves as a natural generalization of standard Euclidean batch normalization. We formalize this fact in the following proposition.

**Proposition D.1.** *The LieBN algorithm presented in Alg. 1 is equivalent to the standard Euclidean BN when $\mathcal{M} = \mathbb{R}^n$, both during the training and testing phases.*

*Proof.* The core of this proof lies in the fact that on $\mathbb{R}^n$, (1) the Fréchet mean and variance are reduced to the familiar Euclidean statistics. (2) the calculation of the running mean becomes the weighted arithmetic mean. (3) Eqs. (13) to (15) become Eq. (6); We prove these three points one by one.

As stated in Lou et al. (2020, Prop. G.1 and Cor. G.2), from the view of the product manifold, the element-wise Fréchet mean and variance on $\mathbb{R}^n$ are equivalent to the vector-valued Euclidean variance and mean.

Besides, by similar proof as in Lou et al. (2020, Prop. G.1), the weighted Fréchet mean on $\mathbb{R}^n$ is simplified as the weighted arithmetic average. Therefore, on $\mathbb{R}^n$, the calculation of running statistics in our Alg. 1 becomes the familiar moving average.

Thirdly, on $\mathbb{R}^n$, we know that $L_x(y) = x + y$, $\operatorname{Exp}_x v = x + v$, $\operatorname{Log}_x y = y - x$, and the neutral element is 0. Since statistics, as well as the Euclidean BN, are calculated element-wisely, we can safely consider a single element, *i.e.*,$\mathbb{R}^n = \mathbb{R}$. For a batch of activations $\{x_{i...N} \in \mathbb{R}\}$, where the batch mean and batch variance are denoted as $\mu_b$ and $v_b^2$, then Eqs. (13) to (15) are rewrote as:

$$L_\beta \left( \operatorname{Exp}_0 \left[ \frac{\gamma}{\sqrt{v_b^2 + \epsilon}} \operatorname{Log}_0(L_{-\mu_b}(x_i)) \right] \right) = \gamma \frac{x_i - \mu_b}{\sqrt{v_b^2 + \epsilon}} + \beta. \tag{39}$$

The above equation is the exact core computation of the standard Euclidean BN.    $\square$

## E    Domain-specific Momentum LieBN for EEG Classification

Kobler et al. (2022a) proposed SPD domain-specific momentum batch normalization (SPDDSMBN) as a domain adaptation approach for EEG classification. SPDDSMBN, based on Eq. (11), performed normalization of mean and variance on SPD manifolds under the specific AIM. Additionally, SPDDSMBN utilized separate momentums for updating training and testing running statistics, inspired by the work of Yong et al. (2020). Following Kobler et al. (2022a, Alg. 1), we also present a momentum LieBN (MLieBN) in Alg. 2. Here $\gamma$ is fixed and $\gamma_{train}$ is defined as

$$\gamma_{train} = 1 - \rho^{\frac{1}{K-1} \max(K-k,0)} + \rho, \text{ where } \rho = \frac{1}{domains\_per\_batch} \tag{40}$$

Furthermore, in line with Kobler et al. (2022a), we adopt multi-channel mechanisms for domain-specific MLieBN (DSMLieBN), where each domain has its own MLieBN layer. Similar to Kobler et al. (2022a), we set the biasing parameter equal to the neutral element, and the scaling factor is shared across all domains. We denote Alg. 2 as $\operatorname{MLieBN}(P_j|M, s, \epsilon, \gamma, \gamma_{train})$. Then our DSM-LieBN follows

$$\operatorname{DSMLieBN}(P_j, i) = \operatorname{MLieBN}_i(P_j|E, s, \epsilon, \gamma, \gamma_{train}), \forall P_j \in \{P_{1...N}\}, \tag{41}$$

---

**Algorithm 2:** Momentum LieBN (MLieBN) Algorithm

---

**Input**  : A batch of activations $\{P_{1...N} \in \mathcal{M}\}$, and a small positive constant $\epsilon$
     running mean $\bar{M}_r = E$, running variance $\bar{v}_r^2 = 1$ for training
     running mean $\tilde{M}_r = E$, running variance $\tilde{v}_r^2 = 1$ for testing
     biasing parameter $B \in \mathcal{M}$, scaling parameter $s \in \mathbb{R}/\{0\}$,
     momentum for training and testing $\gamma_{train}, \gamma \in [0, 1]$
**Output** : Normalized activations $\{\tilde{P}_{1...N}\}$

---

**if** *training* **then**
 Compute batch mean $M_b$ and variance $v_b^2$ of $\{P_{1...N}\}$;
 $\bar{M}_r \leftarrow \text{WFM}(\{1 - \gamma_{train}, \gamma_{train}\}, \{\bar{M}_r, M_b\})$;
 $\bar{v}_r^2 \leftarrow (1 - \gamma_{train})\bar{v}_r^2 + \gamma_{train}v_b^2$;
 $\tilde{M}_r \leftarrow \text{WFM}(\{1 - \gamma, \gamma\}, \{\tilde{M}_r, M_b\})$;
 $\tilde{v}_r^2 \leftarrow (1 - \gamma)\tilde{v}_r^2 + \gamma v_b^2$;
**end**
**if** *training* **then** $M \leftarrow \bar{M}_r, v^2 \leftarrow \bar{v}_r^2$;
**else** $M \leftarrow \tilde{M}_r, v^2 \leftarrow \tilde{v}_r^2$;
**for** $i \leftarrow 1$ **to** $N$ **do**
 Centering to the neutral element $E$: $\bar{P}_i \leftarrow L_{M_{\odot}^{-1}}(P_i)$
 Scaling the dispersion: $\hat{P}_i \leftarrow \text{Exp}_E \left[ \frac{s}{\sqrt{v^2 + \epsilon}} \text{Log}_E(\bar{P}_i) \right]$
 Biasing towards parameter $B$: $\tilde{P}_i \leftarrow L_B(\hat{P}_i)$
**end**

---

where $i$ is the index of the domain. We follow the official code of SPDDSMBN[5] to implement our DSMLieBN. In a word, the only difference between DSMLieBN and SPDDSMBN is the different way of normalization.

Analogous to Thm. 5.3, computations for DSMLieBN under pullback metrics can also be performed by mapping, calculating, and then remapping.

## F BACKPROPAGATION OF MATRIX FUNCTIONS

Our implementation of LieBN on SPD manifolds involves several matrix functions. Thus, we employ matrix backpropagation (BP) (Ionescu et al., 2015) for gradient computation. These matrix operations can be divided into Cholesky decomposition and the functions based on Eigendecomposition.

The differentiation of the Cholesky decomposition can be found in Murray (2016, Eq. 8) or Lin (2019, Props. 4). Besides, our homemade BP of the Cholesky decomposition yields a similar gradient to the one generated by autograd of `torch.linalg.cholesky`. Therefore, during the experiments, we use `torch.linalg.cholesky`.

The second type of matrix functions is based on Eigendecomposition, such as matrix exponential, logarithm, and power. Although torch (Paszke et al., 2019) supports autograd of Eigendecomposition, it requires the computation of $\frac{1}{\delta_i - \delta_j}$ (Ionescu et al., 2015, Props. 1), where $\delta_i$ and $\delta_j$ denote eigenvalues. This might trigger numerical instability when $\delta_i$ approximates $\delta_j$. Following Brooks et al. (2019b), we use the Daleckiĭ-Kreĭn formula (Bhatia, 2013, Thm. V.3.3) to calculate the BP of Eigen-based matrix functions. In detail, for a matrix function defined as $X = f(S) = Uf(\Sigma)U^{\top}$, with $S = U\Sigma U^{\top}$ as the eigendecomposition of an SPD matrix, its BP is expressed as

$$\nabla_S L = U[K \odot (U^T(\nabla_X L)U)]U^T. \tag{42}$$

where $\nabla_X L$ is the Euclidean gradient of the loss function $L$ w.r.t. $X$. Matrix $K$ is defined as

$$K_{ij} = \begin{cases} \frac{f(\sigma_i) - f(\sigma_j)}{\sigma_i - \sigma_j} & \text{if } \sigma_i \neq \sigma_j \\ f'(\sigma_i) & \text{otherwise} \end{cases} \tag{43}$$

---

[5]https://github.com/rkobler/TSMNet

where $\Sigma = \mathrm{diag}(\sigma_1, \sigma_2, \cdots, \sigma_d)$. Eq. (43) demonstrates the numerical stability of Daleckiĭ-Kreĭn formula.

# G   ADDITIONAL DETAILS AND EXPERIMENTS OF LieBN ON SPD MANIFOLDS

We use the official code of SPDNetBN[6] (Brooks et al., 2019b) and TSMNet[7] (Kobler et al., 2022a) to implement our experiments on the SPDNet and TSMNet backbones.

## G.1   DATASETS AND PREPROCESSING

**Radar** dataset (Brooks et al., 2019b) contains 3,000 synthetic radar signals. Following the protocol in Brooks et al. (2019b), each signal is split into windows of length 20, resulting in 3,000 covariance matrices of the size $20 \times 20$ equally distributed in 3 classes. **HDM05** dataset (Müller et al., 2007) consists of 2,273 skeleton-based motion capture sequences executed by different actors. Each frame consists of 3D coordinates of 31 joints, allowing the representation of each sequence as a $93 \times 93$ covariance matrix. In line with Brooks et al. (2019b), we trim the dataset down to 2086 instances scattered throughout 117 classes by removing some under-represented clips. **FPHA** (Garcia-Hernando et al., 2018) includes 1,175 skeleton-based first-person hand gesture videos of 45 different categories with 600 clips for training and 575 for testing. Following Wang et al. (2021), we represent each sequence as a $63 \times 63$ covariance matrix. **Hinss2021** dataset (Hinss et al., 2021) is a recently released competition dataset containing EEG signals for mental workload estimation. The dataset is employed for two tasks, namely inter-session and inter-subject, which are treated as domain adaptation problems. Geometry-aware methods (Yair et al., 2019; Kobler et al., 2022a) have demonstrated promising performance in EEG classification. We follow Kobler et al. (2022a) for data preprocessing. In detail, the python package MOABB (Jayaram & Barachant, 2018) and MNE (Gramfort, 2013) are used to preprocess the datasets. The applied steps include resampling the EEG signals to 250/256 Hz, applying temporal filters to extract oscillatory EEG activity in the 4 to 36 Hz range, extracting short segments ($\leq$ 3s) associated with a class label, and finally obtaining $40 \times 40$ SPD covariance matrices.

## G.2   HYPER-PARAMETERS

We implement the SPD LieBN and DSMLieBN induced by three standard left-invariant metrics, namely AIM, LEM, and LCM, along with their parameterized metrics. Therefore, our method has a maximum of three hyper-parameters, *i.e.,*$(\theta, \alpha, \beta)$, where $\theta$ controls deformation. In our LieBN, $(\alpha, \beta)$ only affects variance calculation. Therefore, we set $(\alpha, \beta) = (1, 0)$ and only tune the deformation factor $\theta$ from the candidate values of $\pm 0.5$, $\pm 1$, and $\pm 1.5$. We denote [Baseline]+[BN_Type]+[Metric]-[$\theta$] as the baseline endowed with a specific LieBN, such as SPDNet+LieBN-AIM-(1) and TSMNet+DSMLieBN-LCM-(1).

## G.3   EVALUATION METHODS

In line with the previous work (Brooks et al., 2019b; Kobler et al., 2022a), we use accuracy as the scoring metric for the Radar, HDM05, and FPHA datasets, and balanced accuracy (*i.e.,*the average recall across classes) for the Hinss2021 dataset. Ten-fold experiments on the Radar, HDM05, and FPHA datasets are carried out with randomized initialization and split (split is officially fixed for the FPHA dataset), while on the Hinss2021 dataset, models are fit and evaluated with a randomized leave 5% of the sessions (inter-session) or subjects (inter-subject) out cross-validation scheme.

## G.4   EMPIRICAL INSIGHTS ON THE HYPER-PARAMETERS IN LieBN ON SPD MANIFOLDS

Our SPD LieBN has at most three types of hyper-parameters: Riemannian metric, deformation factor $\theta$, and $O(n)$-invariance parameters $(\alpha, \beta)$. The general order of importance should be Riemannian metric $> \theta > (\alpha, \beta)$.

The most significant parameter is the choice of Riemannian metric, as all the geometric properties are sourced from a metric. A safe choice would start with AIM, and then decide whether to explore other metrics further. The most important reason is the property of affine invariance of AIM, which

---

[6]https://proceedings.neurips.cc/paper_files/paper/2019/file/6e69ebbfad976d4637bb4b39de261bf7-Supplemental.zip

[7]https://github.com/rkobler/TSMNet

is a natural characteristic of covariance matrices. In our experiments, the LieBN-AIM generally achieves the best performance. However, AIM is not always the best metric. As shown in Tab. 4b, the best result on the HDM05 dataset is achieved by LCM-based LieBN, which improves the vanilla SPDNet by 11.71%. Therefore, when choosing Riemannian metrics on SPD manifolds, a safe choice would start with AIM and extend to other metrics. Besides, if efficiency is an important factor, one should first consider LCM, as it is the most efficient one.

The second one is the deformation factor $\theta$. As we discussed in Sec. 5.1, $\theta$ interpolates between different types of metrics ($\theta = 1$ and $\theta \to 0$). Inspired by this, we select $\theta$ around its deformation boundaries (1 and 0). In this paper we roughly select $\theta$ from $\{\pm 0.5, \pm 1, \pm 1.5\}$

The less important parameters are $(\alpha, \beta)$. Recalling Alg. 1. and Tab. 1, $(\alpha, \beta)$ only affects the calculation of variance, which should have less effects compared with the above two parameters. Therefore, we simply set $(\alpha, \beta) = (1, 0)$ during experiments.

### G.4.1 The Effect of $\beta$ in SPD LieBN

Recalling Eq. (3), $\beta$ controls the relative importance of the trace part against the inner product. Therefore, we set the candidate values of $\beta$ as $\{1, 1/n, 1/n^2, 0, -1/n + \epsilon, -1/n^2\}$, where $n$ is the input dimension of LieBN, and $\epsilon$ is a small positive scalar to ensure $\mathrm{O}(n)$-invariance, *i.e.,* $(\alpha, \beta) \in \mathbf{ST}$. $1/n^2$ and $1/n$ means averaging the trace in Eq. (3), while the sign of $\beta$ denotes suppressing (-), enhancing (+), or neutralizing (0) the trace.

We focus on AIM-based LieBN on the HDM05 dataset. We set $\theta = 1.5$, as it is the best deformation factor under this scenario. Other network settings remain the same as the main paper. The 10-fold average results are presented in Tab. 7. Note that on this setting, $n = 30$. As expected, $\beta$ has minor effects on our LieBN.

Table 7: The effect of different $\beta$ for AIM-based LieBN on the HDM05 dataset.

| Beta | $-1/30^2$ | -0.03 | $1/30^2$ | $1/30$ | 1 | 0 |
|---|---|---|---|---|---|---|
| Mean±STD | 68.18±0.86 | 68.12±0.74 | 68.20±0.85 | 68.18±0.85 | 68.16±0.80 | 68.16±0.68 |

## H Preliminary Experiments on Rotation Matrices

This section implements our LieBN in Alg. 1 on the special orthogonal groups, *i.e.,* $\mathrm{SO}(n)$, also known as rotation matrices. We apply our LieBN to the classic LieNet (Huang & Van Gool, 2017), where the latent space is the special orthogonal group.

### H.1 Geometry on Rotation Matrices

Table 8: The associated Riemannian operators on Rotation matrices.

| Operators | $\mathrm{d}^2(R, S)$ | $\mathrm{Log}_I R$ | $\mathrm{Exp}_I(A)$ | $\gamma_{(R,S)}(t)$ | FM |
|---|---|---|---|---|---|
| Expression | $\left\| \mathrm{mlog}\left(R^\top S\right) \right\|_{\mathrm{F}}^2$ | $\mathrm{mlog}(R)$ | $\mathrm{mexp}(A)$ | $R \, \mathrm{mexp}(t \, \mathrm{mlog}(R^\top S))$ | Manton (2004, Alg. 1) |

We denote $R, S \in \mathrm{SO}(n)$, and $\gamma_{(R,S)}(t)$ as the geodesic connecting $R$ and $S$. The neutral elements of rotation matrices is the identity matrix. Tab. 8 summarizes all the necessary Riemannian ingredients of the invariant metric on $\mathrm{SO}(n)$.

For the specific $\mathrm{SO}(3)$, the matrix logarithm and exponentiation can be calculated without decomposition (Murray et al., 2017, Exs. A. 11 and A.14).

### H.2 Datasets and Preprocessing

Following LieNet, we validate our LieBN on the **G3D** dataset (Bloom et al., 2012). This dataset (Bloom et al., 2012) consists of 663 sequences of 20 different gaming actions. Each sequence is recorded by 3D locations of 20 joints (i.e., 19 bones). Following Huang & Van Gool (2017), we use the code of Vemulapalli et al. (2014) to represent each skeleton sequence as a point on the Lie group $\mathrm{SO}^{N \times T}(3)$, where $N$ and $T$ denote spatial and temporal dimensions. As preprocessed in Huang & Van Gool (2017), we set $T$ as 100 for each sequence on the G3D.

### H.3 IMPLEMENTATION DETAILS

**LieNet:** The LieNet consists of three basic layers: RotMap, RotPooling, and LogMap layers. The RotMap mimics the convolutional layer, while the RotPooling extends the pooling layers to rotation matrices. The logMap layer maps the rotation matrix into the tangent space at the identity for classification. Note that the official code of LieNet[8] is developed by Matlab. We follow the open-sourced Pytorch code[9] to implement our experiments. To reproduce LieNet more faithfully, we made the following modifications to this Pytorch code. We re-code the LogMap and RotPooling layers to make them consistent with the official Matlab implementation. In addition, we also extend the existing Riemannian optimization package geoopt Bécigneul & Ganea (2018) into $SO(3)$ to allow for Riemannian version of SGD, ADAM, and AMSGrad on $SO(3)$, which is missing in the current package. However, we find that SGD is the best optimizer for LieNet. Therefore, we adopt SGD during the experiments. We apply our LieBN before the LogMap layer and refer to this network as LieNetLieBN. Note that the dimension of features in LieNet is $B \times N \times T \times 3 \times 3$, we calculate Lie group statistics along the batch and spatial dimensions ($B \times T$), resulting in an $N \times 3 \times 3$ running mean.

**Training Details:** Following Huang et al. (2017), we focus on the suggested 3Blocks architecture for the G3D dataset. The learning rate is $1e^{-2}$ with a weight decay of $1e^{-5}$. Following LieNet, we adopt a 10-fold cross-subject test setting, where half of the subjects are used for training and the other half are employed for testing.

### H.4 RESULTS

The 10-fold results are shown in Tab. 9. Due to different software, our reimplemented LieNet is slightly worse than the performance reported in Huang et al. (2017). However, we still can observe a clear improvement of LieNetLieBN over LieNet.

Table 9: Results of LieNet with or without LieBN on the G3D dataset.

| Methods | G3D | |
| --- | --- | --- |
| | Mean±STD | Max |
| LieNet | 87.91±0.90 | 89.73 |
| LieNetLieBN | 88.88±1.62 | 90.67 |

## I   PROOFS OF THE LEMMAS AND THEORIES IN THE MAIN PAPER

*Proof of Prop. 4.1 .* Property 1:

The MLE of $M$ is

$$
\begin{aligned}
M_{\mathrm{MLE}} &= \operatorname{argmax} \log(K(v)) - \sum_{i=1}^{N} \frac{\mathrm{d}(P_i, M)^2}{2v^2} \\
&= \operatorname{argmin} \sum_{i=1}^{N} \mathrm{d}(P_i, M)^2.
\end{aligned}
\tag{44}
$$

Property 2:

We denote $Y = L_B(X)$, and $p_X$ and $p_Y$ as the density of $X$ and $Y$, respectively. The density of $Y$ is

$$
\begin{aligned}
p_Y(Q) &= p_X(L_{B_{\odot}^{-1}}(Q)) \\
&= k(\sigma) \exp\left( -\frac{\mathrm{d}(L_{B_{\odot}^{-1}}(Q), M)^2}{2\sigma^2} \right) \\
&= k(\sigma) \exp\left( -\frac{\mathrm{d}(Q, L_B(M))^2}{2\sigma^2} \right).
\end{aligned}
\tag{45}
$$

---

[8]https://github.com/zhiwu-huang/LieNet
[9]https://github.com/hjf1997/LieNet

The first equation is obtained by (Pennec, 2004, Thm. 7), while the last equation is obtained by the isometry of the left translation. □

*Proof of Prop. 4.2* . The isometry of $L_B$ can directly obtain the homogeneity of the sample mean. Now let us focus on Eq. (17). We have the following:

$$
\begin{aligned}
\sum_{i=1}^{N} w_i \, \mathrm{d}^2(\phi_s(P_i), E) &= \sum_{i=1}^{N} w_i \| s \operatorname{Log}_E P_i \|_E \\
&= s^2 \sum_{i=1}^{N} w_i \| \operatorname{Log}_E P_i \|_E^2 \\
&= s^2 \sum_{i=1}^{N} w_i \, \mathrm{d}^2(P_i, E),
\end{aligned}
\tag{46}
$$

where $\| \cdot \|_E$ is the norm on $T_E \mathcal{M}$. □

*Proof of Prop. 5.1* . We first prove the case of $(\theta, \alpha, \beta)$-LEM, and then proceed to the case of $\theta$-LCM.

$(\theta, \alpha, \beta)$-**LEM:** For clarity, we denote the metric tensor of $(\theta, \alpha, \beta)$-LEM as

$$
g^{(\theta,\alpha,\beta)\text{-LE}} = \frac{1}{\theta^2} \mathrm{P}_\theta^* \, g^{(\alpha,\beta)\text{-LE}},
\tag{47}
$$

where $g^{(\alpha,\beta)\text{-LE}}$ is the metric tensor of $(\alpha, \beta)$-LEM. Let $P \in \mathcal{S}_{++}^n$ and $V, W \in T_P \mathcal{S}_{++}^n$, then we have

$$
\begin{aligned}
g_P^{(\theta,\alpha,\beta)\text{-LE}}(V, W) &= \frac{1}{\theta^2} g_{\mathrm{P}_\theta(P)}^{(\alpha,\beta)\text{-LE}} \left( \mathrm{P}_{\theta*,P}(V), \mathrm{P}_{\theta*,P}(W) \right) \\
&= \frac{1}{\theta^2} \langle (\mathrm{mlog} \circ \mathrm{P}_\theta)_{*,P}(V), (\mathrm{mlog} \circ \mathrm{P}_\theta)_{*,P}(W) \rangle^{(\alpha,\beta)} \\
&= \langle \mathrm{mlog}_{*,P}(V), \mathrm{mlog}_{*,P}(W) \rangle^{(\alpha,\beta)} \\
&= g_P^{(\alpha,\beta)\text{-LE}}(V, W).
\end{aligned}
\tag{48}
$$

$\theta$-**LCM:** Let us first review a well-known fact of deformed metrics (Thanwerdas & Pennec, 2022a). Let $\tilde{g} = \frac{1}{\theta^2} \mathrm{P}_\theta^* g$ be the power-deformed metric on SPD Then when $\theta$ tends to 0, for all $P \in \mathcal{S}_{++}^n$ and all $V \in T_P \mathcal{S}_{++}^n$, we have

$$
\tilde{g}_P(V, V) \to g_I(\log_{*,P}(V), \log_{*,P}(V)).
\tag{49}
$$

By Eq. (49), we can readily obtain the results. □

*Proof of Prop. 5.2* . $(\alpha, \beta)$-AIM is left-invariant (Thanwerdas & Pennec, 2022b). As the pullback of $(\alpha, \beta)$-AIM, $(\theta, \alpha, \beta)$-AIM is left-invariant as well. Besides, Chen et al. (2023d) shows that LCM is the pullback metric from the Euclidean space of $\mathcal{L}^n$. Therefore, $\theta$-LCM is bi-invariant. □

*Proof of Thm. 5.3* . We denote Eqs. (13) to (15) on $\mathcal{M}_i, i = 1, 2$ as the mapping $\xi^i(\cdot | M, v^2, B, s)$. Let $\mathcal{B} = \{P_{1\ldots N}\}$ and $f(\mathcal{B}) = \{f(P_{1\ldots N})\}$.

The core of this proof lies in three points:

1. The Fréchet mean and variance of $\mathcal{B}$ in $\mathcal{M}_1$ correspond to the counterparts of $f(\mathcal{B})$ in $\mathcal{M}_2$.

2. $\xi^1(P_i | M, v^2, B, s)$ in $\mathcal{M}_1$ is equal to $f^{-1}(\xi^2(f(P_i) | f(M), v^2, f(B), s))$.

3. The updates of running statistics in $\mathcal{M}_1$ correspond to the counterparts in $\mathcal{M}_2$.

We denote $M$ as the Fréchet mean of $\mathcal{B}$, and $v^2$ as the Fréchet variance of $\mathcal{B}$. Then, by the isometry of $f$, the Fréchet mean and variance of $f(\mathcal{B})$ are $f(M)$ and $v^2$, respectively.

On $\mathcal{M}_i, i = 1, 2$, we denote $L^i, \odot^i, \operatorname{Exp}^i, \operatorname{Log}^i$ as the Lie group and Riemannian operators, $E^i$ as the neutral element, and Eq. (14) as $\phi_s^i(\cdot)$. With the isometry and Lie group isomorphism of $f$, we

have the following equations:

$$L^1_{M^{-1}_{\odot^1}} = f^{-1} \circ L^2_{f(M)^{-1}_{\odot^2}} \circ f, \tag{50}$$

$$\begin{aligned}
\phi^1_s &= \mathrm{Exp}^1_{E^1}\left[s\,\mathrm{Log}^1_{E^1}(\cdot)\right] \\
&= f^{-1}\left(\mathrm{Exp}^1_{E^2}\left[s\,\mathrm{Log}^2_{E^2}(f(\cdot))\right]\right) \\
&= f^{-1} \circ \phi^2_s \circ f, \tag{51}
\end{aligned}$$

$$L^1_B = f^{-1} \circ L^2_{f(B)} \circ f. \tag{52}$$

Then we have

$$\xi^1(P_i|M, v^2, B, s) = f^{-1}(\xi^2(f(P_i)|f(M), v^2, f(B), s)) \tag{53}$$

Lastly, we show the correspondence between running statistics. Since the Fréchet variance is the same for both $\mathcal{B}$ and $f(\mathcal{B})$, we focus on the running mean. Let $M_r$ and $f(M_r)$ denote the initial values of the running means in $\mathcal{M}_1$ and $\mathcal{M}_2$ respectively, and $\mathrm{WFM}^i$ represent the weighted Fréchet mean in $\mathcal{M}_i$. Then the updated running mean in $\mathcal{M}_1$ is

$$\mathrm{WFM}^1(\{1-\gamma, \gamma\}, \{M_r, M\}) = f^{-1}(\mathrm{WFM}^2(\{1-\gamma, \gamma\}, \{f(M_r), f(M)\})) \tag{54}$$

we can further simply the above equation as

$$\mathrm{WFM}^1 = f^{-1} \circ \mathrm{WFM}^2 \circ f \tag{55}$$

Denoting $\mathrm{LieBN}^i$ as the LieBN algorithm on $\mathcal{M}_i$, Eq. (53) and Eq. (55) imply that:

$$\mathrm{LieBN}^1(P_i|B, s, \epsilon, \gamma) = f^{-1}\left[\mathrm{LieBN}^2(f(P_i)|f(B), s, \epsilon, \gamma)\right]. \tag{56}$$

$\square$

