# OpenReview forum: "A Lie Group Approach to Riemannian Batch Normalization"
_ICLR.cc/2024/Conference — ICLR 2024 poster_

### Official Review · Reviewer_R6Kt · 2023-10-24

**Soundness:** 3 good
**Presentation:** 3 good
**Contribution:** 3 good
**Rating:** 6
**Confidence:** 3

**Summary:**

This paper proposes a batch-normalization (BN) method for manifold-valued features in neural networks. While in prior works several BN techniques are proposed for specific types of manifold, the authors present a general manifold-based BN formulation from a viewpoint of Lie-group. Besides, especially for SPD manifolds, practical BN methods are derived from the general formulation based on three types of pull-back metrics [Chen+23].
The experimental results using SPDnet and TSMnet demonstrate that the propose methods exhibit competitive performance to SOTAs.

**Strengths:**

+ A general formulation of manifold-based BN is presented through reviewing/summarizing several Riemannian-normalization (RN) approaches.
+ Practical BNs for SPD matrices are derived in an efficient form from the general formulation.

**Weaknesses:**

Novelty of LieBN in Sec.4 is limited as it is rather straightforward from the prior works [Kobler+22a] and [Chakraborty+20].

On the other hand, the pull-back metrics [Chen+23ab] are effectively applied to the general formulation to instantiate practical BN methods for SPD manifolds in an interesting way.
Though especially pull-back Euclidean metrics seem to be efficient as shown in Table 3, this paper lacks in-depth analysis about the methods from qualitative and/or computational viewpoint.
It is demanded to clarify computational details such as by showing back-props through comparison to the other RN approaches based on complicated manifold-based computation, which would significantly improve reproducibility.

Considering \theta-parameterization does not work so well as shown in Sec.6, such a parametric extension might be redundant, rather complicating the discussion in Sec.4. In stead of that extension, it may be better to focus on analyzing the practical BN methods shown in Table 3.

As to empirical performance results reported in the experiments, superiority of the method is less clear since the performance improvement is not significant due to large stds of performance scores.
To clarify the efficacy of the proposed method, it should be compared with the other RN methods in terms of computation cost not only the classification performance.

Based on the experimental results, one cannot identify the best SPD-BN method that outperforms the others consistently. Although the authors insist such an inconsistency shows generality of the approach, it is less understandable and unfavorable from a practical viewpoint. In this case, provide some discussion and/or analysis about connection between types of metrics and tasks (or network architectures) for rendering insights into the SPD metrics.

Minor comments:
In Eq.5: $\frac{1}{N} \sum$ -> $\sum$

**Questions:**

The above-mentioned concerns should be addressed especially in the following points.
- Analysis about the SPD-BN methods in Table 3 from computational viewpoint in comparison to the other RNs.
- Empirical comparison regarding computation cost.

---

> ### Author Response · Authors · 2023-11-17
> **Response to Reviewer R6Kt (R4)**
>
> We thank Reviewer $\textcolor{blue}{R6Kt}$ ($\textcolor{blue}{R4}$) for the thoughtful comments. We respond to the concerns as follows.
> ***
>
> **1. Difference with the prior work [Kobler+22a] and [Chakraborty+20].**
>
> The theory and implementation of our LieBN are different from [Kobler+22a] and more general than [Chakraborty+20]. Please refer to CQ#1 in the common response.
>
> **2. Computational details on back-props**
>
> As summarized in the following table, two types of matrix functions are involved in our LieBN on SPD manifolds: one is based on Eigendecomposition, and the other is the Cholesky decomposition. The Eigen-based matrix functions include matrix logarithm, exponential, and power. Their BP can be calculated by the  Daleckii-Krein formula [2 Thm. V.3.3]. The BP of the Cholesky decomposition has been well established in `torch.linalg.cholesky`. Therefore, we use `torch.linalg.cholesky` for the Cholesky decomposition. We have added a detailed discussion on matrix BP in the App. F in our revised manuscript.
>
> Table A: Matrix functions involved in LieBN on SPD manifolds
> |       Matrix Function      |          Type          |       Involvement       |
> |:--------------------------:|:----------------------:|:-----------------------:|
> |      Matrix Logarithm      |        Eigen-based       | LieBN-AIM and LieBN-LEM |
> |     Matrix Exponential     |        Eigen-based       | LieBN-AIM and LieBN-LEM |
> |     Matrix Square Root     |        Eigen-based       |        LieBN-AIM        |
> | Matrix Inverse Square Root |        Eigen-based       |        LieBN-AIM        |
> |        Matrix Power        |        Eigen-based       |        LieBN-AIM and power deformation        |
> |   Cholesky Decomposition   | Cholesky Decomposition |        LieBN-LCM        |
>
> **3. Large std on EEG**
>
> Recalling Tab. 4-5, it is only on the EEG datasets that the performance shows relatively large std, which should be attributed to the characteristics of the EEG dataset. EEG signals exhibit low signal-to-noise ratio (SNR), domain shifts, and low specificity. These challenges are more obvious in the inter-session and inter-subject scenarios.
>
> Despite the relatively large std, our LieBN outperforms the SPDDSMBN on the inter-subject classification by 3.87% (50.10 v.s. 53.97). Besides, our LieBN can show less std than the SPDDSMBN baseline on EEG datasets. For instance, for the inter-subject task (Tab. 5 (b)), the std of LCM-based LieBN is clearly less than SPDDSMBN ( 51.53±4.96 v.s. 50.10±8.08).
>
> **4.  Discussion and analysis about the choice of metrics.**
>
> The foremost challenge when building Riemannian learning algorithms is choosing an appropriate Riemannian metric. If the framework is general, there would be many metrics to select from, and the model would be more likely to perform better. This is why we view the generality of our framework as an advantage.
>
> Generally speaking, AIM is the best candidate metric on SPD manifolds. The most significant reason is the property of affine invariance, which is a natural characteristic of describing covariance matrices. In our experiments, the LieBN-AIM generally achieves the best performance. In some scenarios, there are better choices than AIM. As shown in Tab. 4 (b), the best result on the HDM05 dataset is achieved by LCM-based LieBN, which improves the vanilla SPDNet by 11.71%. Therefore, when choosing Riemannian metrics on SPD manifolds, a safe choice would start with AIM and extend to other metrics.

---

> > ### Author Response · Authors · 2023-11-17
> > **Computational analysis and comparison**
> >
> > **5. Computational analysis and comparison.**
> >
> > Our paper compares our LieBN against two existing SPDBNs, including SPDNetBN and SPDDSMBN. As SPDDSMBN is the improved version of SPDNetBN, let us focus on comparing our LieBN and SPDDSMBN. For simplicity, we focus on LieBN induced by the standard ($\theta=1$) AIM, LEM, and LCM.
> >
> > **Computational comparison of LieBNs under different metrics**
> > Recalling Tab. 3, the LCM-based LieBN only contains two Cholesky decompositions, while LEM-based LieBN contains two eigendecompositions. Note that Cholesky decomposition is more efficient than eigendecomposition. On the other hand, the AIM-based LieBN involves several more matrix decompositions. Therefore, our LieBN's efficiency order should be LieBN-LCM > LieBN-LEM > LieBN-AIM.
> >
> > **Computational comparison of LieBNs against SPDDSMBN**
> >
> > As SPDDSMBN is based on AIM, it should be less efficient than our LieBN-LCM and LieBN-LCM. Besides, the major difference between our LieBN-AIM and SPDDSMBN lies in biasing and centering. Our LieBN-AIM uses Cholesky decomposition for biasing and centering, while SPDDSMBN adopts the matrix square root. As the Cholesky decomposition is more efficient than the matrix square root, we could expect our LieBN-AIM to be more efficient than SPDDSMBN.
> >
> > **Empirical training cost**
> > The general order of efficiency should be LieBN-LCM > LieBN-LEM > LieBN-AIM>SPDDSMBN. This can be demonstrated by the average training time per epoch presented in Tab. 4 - Tab.5. Tab. 5 (b) shows that our LieBN is generally much more efficient than SPDDSMBN. In detail, LieBN-LCM only takes half of the training time of SPDDSMBN (7.74 v.s. 3.59). LieBN-LEM is also much more efficient than SPDDSMBN (7.74 v.s. 4.71). Besides, LieBN-AIM also achieves better efficiency than SPDDSMBN (7.74 v.s. 6.94).
> >
> > >[1] Ionescu C, Vantzos O, Sminchisescu C. Matrix backpropagation for deep networks with structured layers.
> > >
> > >[2] Bhatia R. Matrix analysis

---

> > > ### Comment · Reviewer_R6Kt · 2023-11-22
> > >
> > > Thanks for the response.
> > > It clarifies my concern especially regarding the computation issues.

---

### Official Review · Reviewer_LaGM · 2023-10-24

**Soundness:** 3 good
**Presentation:** 2 fair
**Contribution:** 1 poor
**Rating:** 3
**Confidence:** 3

**Summary:**

The paper concerns batch normalization for Lie group valued data. The authors propose a unified framework for batch normalization that they claim offers theoretical guarantees.

**Strengths:**

I have a hard time finding strengths that were not already presented in previous papers. I hope the authors can argue to the opposite, but as of now I am not sure of what is the actual contribution of the paper.

**Weaknesses:**

- I am unsure what is the contribution of the paper. As the authors state, the normalization scheme they propose has been used in previous work. There are some claims like "In contrast, our work provides a more extensive examination, encompassing both population and sample properties of our LieBN in a general manner. Besides, all the discussion about our LieBN can be readily transferred to right-invariant metrics. " but I was not able to find out what specifically these differences are. The approach seems to be almost exactly the same when I look up in the cited papers where it is applied to Lie groups as well.
- using the Riemannian or Lie group exp and log maps for batch normalization was a good idea the first time it was presented, but I don't see the value added with the current paper

**Questions:**

I believe the authors need to argue convincingly what is the contribution of the paper, and why the paper presents a significant contribution relative to the previously methods.

---

> ### Author Response · Authors · 2023-11-17
> **Response to Reviewer LaGM (R3)**
>
> We thank reviewer $\textcolor{green}{LaGM}$ ($\textcolor{green}{R3}$) for the valuable comment. In the following, we respond to the concerns in detail.
> ***
>
> **1. Difference from the previous RBN methods.**
>
> The RBNs in these three works are **_either designed for a specific manifold or metric_**, or fail to control mean and variance, while our LieBN fulfills normalization on general Lie groups. We have summarized their limitations in Tab. 2 and discussed the difference in Sec. 3.2. Here we clarify the difference between our LieBN and their RBN in details.
>
> **- Difference with the RBN proposed in [1]**
>
> **_The RBN in [1] is confined within the standard Affine-Invariant Metric (AIM)_**. In contrast, we implement our LieBN under three invariant metrics: Log-Euclidean Metric (LEM), Log-Cholesky Metric (LCM), and AIM. Moreover, our LieBN further covers the deformed families of LEM, LCM, and AIM.
>
> **_For the specific AIM, our LieBN is also better than the RBN [1] in terms of computational efficiency._** The RBN in [1] is based on parallel transportation, Riemannian exp and log. The specific equation is calculated by the matrix power function (Eq. (10) in [1]). In contrast, our LieBN is based on Lie group left translation, which is fulfilled by Cholesky decomposition (Tab. 3). Note that Cholesky decomposition is more efficient than the matrix power function. As shown in Tab. 5 (b), DSMLieBN-AIM-(1) achieves similar performance to SPDDSMBN with less training time (6.94 v.s. 7.74).
>
> **_Besides, the RBN in [1] is problematic when extending to other manifolds or metrics._** As shown in Eq. (9), the core formulation of the RBN in [1] can be described as
> $$
> \Gamma _{P \rightarrow Q}(S)=\operatorname{Exp} _Q\left[\mathrm{PT} _{P \rightarrow Q}\left(\operatorname{Log} _P(S)\right)\right].
> $$
>
> Since $\Gamma_{P \rightarrow Q}$ becomes affine action under AIM, this formulation can coincidentally control mean under AIM on SPD manifolds. However, the resultant mean and variance are generally agnostic under general manifolds. Therefore, the RBN based on $\Gamma _{P \rightarrow Q}$ could not be extended into other metrics or manifolds. On the contrary, our Props. 4.1-4.2 guarantee the rationality of our LieBN on other Lie groups, such as SO(n). For experiments on SO(n), please refer to the 3rd response to $\textcolor{red}{R1}$.
>
> **- Difference with the RBN proposed by [3]**
>
> **_Similar to [1], the RBN in [3] generally fails to normalize mean or variance either_,** as this RBN is a variant of $\Gamma_{P \rightarrow Q}$. In contrast, our LieBN can always control both mean and variance on Lie groups.
>
> **- Difference with the RBN proposed by [4]**
>
> **_The RBN on matrix Lie groups in [4] is also confined within a specific metric and group structures (Sec. 3.2 in [4])_.** On the contrary, our LieBN is designed for general Lie groups. In this sense, the RBN on matrix Lie groups in [4] is a special case of our LieBN. Furthermore, as stated at the end of Sec. 4, our LieBN framework can be extended into right-invariant metrics.
>
> Besides, as stated in Rmk. C.5 in the appendix, the proof presented in [4] is a bit problematic, as the author fails to consider Riemannian volume when dealing with the probability density function. In contrast, our theoretical results in Props. 4.1-4.2 are more solid and general.
>
> **- Summary**
>
> In summary, the RBNs in [1-3] are either confined within a specific manifold or metric, or fail to control mean and variance. In contrast, we propose a  framework of batch normalization over general Lie groups and showcase its generality over three families of Lie groups on SPD manifolds.
>
> > [1] Kobler R, Hirayama J, Zhao Q, et al. SPD domain-specific batch normalization to crack interpretable unsupervised domain adaptation in EEG.
> >
> > [2] Brooks D, Schwander O, Barbaresco F, et al. Riemannian batch normalization for SPD neural networks.
> >
> > [3] Lou A, Katsman I, Jiang Q, et al. Differentiating through the fréchet mean.
> >
> > [4] Chakraborty R. Manifoldnorm: Extending normalizations on Riemannian manifolds.

---

> > ### Comment · Reviewer_LaGM · 2023-11-22
> > **Response**
> >
> > Thank you for the response. The response has not changed my rating.

---

> > > ### Author Response · Authors · 2023-11-22
> > > **Thanks for your feedback!**
> > >
> > > Thanks for your feedback! Our last response clarified our contributions and the main difference with previous RBN methods. The paper has also been modified accordingly. From our perspective, your main question should have been (largely) solved.
> > >
> > > Could you elaborate more on your remaining concerns? We still have time to answer. 😄

---

### Official Review · Reviewer_B3vJ · 2023-10-31

**Soundness:** 4 excellent
**Presentation:** 3 good
**Contribution:** 3 good
**Rating:** 8
**Confidence:** 5

**Summary:**

Study of Deep Neural Networks (DNNs) on manifolds, associated with normalization techniques, with a unified framework for Riemannian Batch Normalization (RBN) techniques on Lie groups. Theoretical guarantee are provided to caracterize the stability of the process.
Approach is illustrated for Symmetric Positive Definite (SPD) manifolds, with three families of parameterized Lie groups, in a SPD neural networks.  Experiments have been done for radar recognition, human action recognition, and electroencephalography (EEG) classification.

**Strengths:**

Interesting algorithm LieBN, which enables batch normalization over Lie groups, to normalize both the sample and population statistics.and apply to SPD manifolds.

**Weaknesses:**

Density of probability on SPD matrix could be only defined as invariant to all the automorphisms of SPD manifold. To assess which density verify this property, you have to consider "Lie Groups Thermodynamics" developped by Jean-Marie Souriau. Consider upper-half space of Siegel (pure imaginary axis is the space of SPD matrix) where the Lie group SU(n,n) acts transitivelly. With Souriau method, you are able to compute the Gibbs density of maximum entropy that is covariant to SU(n,n). If you restrict to the imaginary axis, you find the density for SPD matrices. See the following reference and put it in your references:
[A] Barbaresco, F. (2021). Gaussian Distributions on the Space of Symmetric Positive Definite Matrices from Souriau’s Gibbs State for Siegel Domains by Coadjoint Orbit and Moment Map. In: Nielsen, F., Barbaresco, F. (eds) Geometric Science of Information. GSI 2021. Lecture Notes in Computer Science(), vol 12829. Springer, Cham. https://doi.org/10.1007/978-3-030-80209-7_28

**Questions:**

Add the following references on batch normalization
[B] Daniel Brooks. Deep Learning and Information Geometry for Time-Series Classification. Machine Learning [cs.LG]. Sorbonne Université, 2020. English. ⟨NNT : 2020SORUS276⟩. ⟨tel-03984879⟩; https://theses.hal.science/tel-03984879
[C] D. Brooks, O. Schwander, F. Barbaresco, J. . -Y. Schneider and M. Cord, "Deep Learning and Information Geometry for Drone Micro-Doppler Radar Classification," 2020 IEEE Radar Conference (RadarConf20), Florence, Italy, 2020, pp. 1-6, doi: 10.1109/RadarConf2043947.2020.9266689.
[D] D. Brooks, O. Schwander, F. Barbaresco, J. -Y. Schneider and M. Cord, "A Hermitian Positive Definite neural network for micro-Doppler complex covariance processing," 2019 International Radar Conference (RADAR), Toulon, France, 2019, pp. 1-6, doi: 10.1109/RADAR41533.2019.171277.
[E] D. A. Brooks, O. Schwander, F. Barbaresco, J. -Y. Schneider and M. Cord, "Complex-valued neural networks for fully-temporal micro-Doppler classification," 2019 20th International Radar Symposium (IRS), Ulm, Germany, 2019, pp. 1-10, doi: 10.23919/IRS.2019.8768161.
[F] D. A. Brooks, O. Schwander, F. Barbaresco, J. -Y. Schneider and M. Cord, "Exploring Complex Time-series Representations for Riemannian Machine Learning of Radar Data," ICASSP 2019 - 2019 IEEE International Conference on Acoustics, Speech and Signal Processing (ICASSP), Brighton, UK, 2019, pp. 3672-3676, doi: 10.1109/ICASSP.2019.8683056.
[G] Brooks, D., Schwander, O., Barbaresco, F., Schneider, JY., Cord, M. (2019). Second-Order Networks in PyTorch. In: Nielsen, F., Barbaresco, F. (eds) Geometric Science of Information. GSI 2019. Lecture Notes in Computer Science(), vol 11712. Springer, Cham. https://doi.org/10.1007/978-3-030-26980-7_78

**Details Of Ethics Concerns:**

I have no conflict of interest with authors.

---

> ### Author Response · Authors · 2023-11-17
> **Response to Reviewer B3vJ (R2)**
>
> We thank reviewer $\textcolor{brown}{B3vJ}$ ($\textcolor{brown}{R2}$) for the encouraging feedback and constructive comments! The following is our detailed response.
> ***
>
> **1. Gaussian distribution by Lie groups thermodynamics.**
>
> Thanks for the valuable comment. The Gaussian distribution you mentioned on the SPD manifold is very inspiring. We have added this reference in Sec. 4, where we introduce Gaussian distribution on manifolds. In our future research, we will explore this distribution and attempt to establish statistics learning methods based on this Gaussian distribution.
>
> **2. Adding some reference on Riemannian batch normalization.**
>
> We have added the reference you mentioned in the main paper. These papers are very interesting, especially HPDNet on Hermitant Positive Definite (HPD) manifolds proposed in [1]. As HPD manifolds are the counterparts of SPD manifolds in the complex domain, HPD manifolds also have Lie group structures. Theoretically, our LieBN framework could also be implemented on HPD neural networks. We will explore these possibilities in the future.
>
> > [1] Brooks, D., Schwander, O., Barbaresco, F., Schneider, J. Y., & Cord, M. A Hermitian Positive Definite neural network for micro-Doppler complex covariance processing.

---

### Official Review · Reviewer_ZRPd · 2023-11-01

**Soundness:** 3 good
**Presentation:** 3 good
**Contribution:** 1 poor
**Rating:** 1
**Confidence:** 4

**Summary:**

This paper proposes a batch normalization layer for neural networks on Lie groups. The authors then focus on SPD neural networks to showcase their approach. The proposed method is validated on radar recognition, action recognition, and electroencephalography (EEG) classification.

**Strengths:**

* Proofs are given in the supplementary material (I did not thoroughly check them)
* Experiment results show improvements over some state-of-the-art SPD neural networks

**Weaknesses:**

* The paper lacks of novelty
* Experimental results are not convincing
* No discussion about the limitations of the proposed approach

**Questions:**

The proposed technique is a simple tweak of those from Kobler et al. (2022b), Lou et al. (2020), Chakraborty (2020).
No new concepts or ideas have been developped w.r.t. these works. While the authors state that the proposed technique works for Lie groups and is able to control mean and variance in contrast to these works, extensions of these works to Lie groups, as done in the paper, are trivial.

The experimental results are not convincing since the proposed method is only compared with some SPD neural networks. For example, on human action recognition, the proposed method is outperformed by the method of Laraba et al. (2017) on HDM05 dataset by a large margin (72.27\% vs. 83.33\%). This shows that the proposed technique is probably not effective compared to other learning techniques designed in Euclidean space.

*Question*

How does the proposed method perform on another Lie groups, e.g. when being used in LieNet (Huang et al., 2017) ?

*References*

1. Sohaib Laraba, Mohammed Brahimi, Joëlle Tilmanne, Thierry Dutoit: 3D skeleton-based action recognition by representing motion capture sequences as 2D-RGB images. Comput. Animat. Virtual Worlds 28(3-4) (2017)

2. Zhiwu Huang, Chengde Wan, Thomas Probst, Luc Van Gool: Deep Learning on Lie Groups for Skeleton-Based Action Recognition. CVPR 2017: 1243-1252.

---

> ### Author Response · Authors · 2023-11-17
> **Response to Reviewer ZRPd (R1)**
>
> We thank reviewer $\textcolor{red}{ZRPd}$ ($\textcolor{red}{R1}$) for the valuable feedbacks. Below is our detailed response.
> ***
>
> **1. The proposed technique is not a simple tweak of those from Kobler et al. (2022b), Lou et al. (2020), and Chakraborty (2020).**
>
> As stated in CQ#1 of the common response, our work is entirely theoretically different from Kobler et al. (2022b) and Lou et al. (2020), and more general than Chakraborty (2020). Please refer to the common response for details.
>
> **2. The experimental results are convincing, given the only modification to SPDNet is the BN layer.**
>
> This paper mainly focuses on building LieBN on general Lie groups. **_Since we only change the model by adding a single BN layer, this slight modification is not expected to outperform every baseline._** Nevertheless, our experiments on the SPD manifolds indicate that our LieBN can improve the performance of SPD neural networks. **On the Radar, HDM05, FPHA, and EEG datasets, our LieBN improves the SPD baselines by 2.22%, 11.71%, 4.8% and 3.87%, respectively**. This is convincing to claim that SPD neural networks can benefit from our LieBN.
>
> **_Besides, SPDDSMBN is one of the SOTA methods in EEG classification._** Tab. 5 indicates the superiority of our work against SPDDSMBN. Moreover, Tab. 5 shows that LCM- or LEM-based LieBN can achieve comparable performance against SPDDSMBN with less training time. For instance, for inter-subject classification, LCM-based LieBN reaches similar results to SPDDSMBN while costing only half of the training time. Besides, LCM-based LieBN tends to show smaller std than SPDDSMBN (4.96 v.s. 8.08 for the inter-subject task)
>
> **3 LieBN on LieNet (Huang et al., 2017)**
>
> The Lie group in LieNet [1] is SO(3). Our LieBN can also be implemented in this Lie group. However, we only implement centering and biasing operations for now due to limited time.
>
> **Theoretical ingredient**
>
> Since we only implement the centering and biasing of LieBN, the core operation of LieBN becomes
> $$
> L _{B} \circ L _{M _{\odot} ^{-1}}(P _i) = B M ^{-1} P _i
> $$
> where $\{P _i\}$ is a batch of SO(3), $M$ is the SO(3) batch mean，and $B$ is an SO(3) biasing parameters. The batch mean can be obtained by [2, Alg. 1], while the running mean can be updated by the geodesic connecting the running mean and batch mean.  The calculation of batch mean [2, Alg. 1] requires Lie group exp & log on SO(3). We present all the necessary ingredients in Tab. 1. For more detail, please refer to [2-3].
>
>
> Table 1: Riemannian operators required in LieBN on SO(3)
> |       Operator      |          Expression          |
> |:--------------------------:|:----------------------:|
> |Lie group exponentiation |$\exp(V)$|
> |Lie group logarithm |$\log(V)$| )\log(V)$|
> |Geodesic connecting $S$ and $R$|$\gamma (t;S,R)= S\exp(t\log(S ^\top R))$|
> |Weighted Fréchet mean| [2, Alg. 1]|
>
> **Implementation details and dataset**
>
> As LieNet was originally implemented by Matlab, we use the open-sourced Pytorch code [4] to reimplement LieNet by torch. We found that the matrix logarithm in the Pytorch code [4] fails to deal with singular cases well. Hence, we use Pytorch3D [5] to calculate the matrix logarithm, i.e., `pytorch3d.transforms.matrix_to_axis_angle`. Besides, we implement matrix exponentiation by Rodrigues' formula [6, Eq. (2)].
>
> Following LieNet, we adopt the G3D-Gaming dataset [7], and follow all the learning settings as in the original paper of LieNet. We apply the LieBN layer before each pooling layer. As the RotMap layer in the original LieNet is actually based on left translation as well, we use the LieBN layer to substitute the RotMap layer. Following LieNet, we adopt the architecture of three pooling layers, which can be illustrated as
> $$
> f _{\mathrm{LieBN}} \rightarrow f _{\mathrm{RotPooling}} \rightarrow f _{\mathrm{LieBN}} \rightarrow f _{\mathrm{RotPooling}} \rightarrow f _{\mathrm{LieBN}} \rightarrow f _{\mathrm{RotPooling}} \rightarrow f _{\log} \rightarrow f _{\mathrm{FC}}.
> $$
> where $f _{\log}$ and $f _{\mathrm{FC}}$ denote the matrix logarithm and FC layer. Similar to LieNet, we train the network by the standard cross-entropy loss.
>
> **Results**
>
> The results are presented in the following table. Due to different software, our reimplemented LieNet is slightly worse than the performance reported in [1]. However, we still can observe a clear improvement of LieNetLieBN over LieNet.
>
> Table 2: Results on LieNet with or without LieBN
> |       Methods      |          Result          |
> |:--------------------------:|:----------------------:|
> |LieNet |86.06|
> |LieNetLieBN |88.18|

---

> > ### Author Response · Authors · 2023-11-17
> > **Reference**
> >
> > > [1] Huang Z, Wan C, Probst T, et al. Deep learning on lie groups for skeleton-based action recognition.
> > >
> > > [2] Manton J H. A globally convergent numerical algorithm for computing the centre of mass on compact Lie groups.
> > >
> > > [3] Boumal N, Absil P A. A discrete regression method on manifolds and its application to data on SO (n).
> > >
> > > [4] https://github.com/hjf1997/LieNethttps://github.com/hjf1997/LieNet
> > >
> > > [5] Ravi N, Reizenstein J, Novotny D, et al. Accelerating 3d deep learning with pytorch3d.
> > >
> > > [6] Nurlanov Z. Exploring SO (3) logarithmic map: degeneracies and derivatives.
> > >
> > > [7] Bloom V, Makris D, Argyriou V. G3D: A gaming action dataset and real time action recognition evaluation framework.

---

> ### Comment · Reviewer_ZRPd · 2023-11-22
>
> Thank the authors for the response. I still keep my original rating since I fail to see the significance of this work with respect to existing works.

---

### Official Review · Reviewer_Urfr · 2023-11-19

**Soundness:** 4 excellent
**Presentation:** 3 good
**Contribution:** 3 good
**Rating:** 8
**Confidence:** 5

**Summary:**

The authors describe a new kind of Batch Normalization layer for
Riemaniann neural networks. The proposed technique is a theoretical
improvement over existing batch-norm layers by being a generalized and
unified view on all previously proposed technique, using the Lie-group
structure of the manifold.

**Strengths:**

- A very pleasant to read recap on all batch norm-like layers for
  Riemannian networks.
- Theoretical guaranties on the control provided by the layer.
- Convincing experimental evaluation (not SOTA obviously, but an
  improvement over other manifold methods)
- LieBN reduces to the classical BN for Euclidean manifold.

**Weaknesses:**

- Nearly all tables are barely readables.
- Novelty of the work should emphasized more. LieBN provides more
  guaranties and a more sound approach. But in the writing, it is not
  completely clear of what is a full novelty over previous methods and
  what is a generalization.
- A broad zoo of choices (AIM, LEM, LCM and alpha beta variants), but no
  clue on choosing. It's an usual question with this type of methods,
  and it always a little bit disapointing to simply benchmark over all
  the possible choices.

**Questions:**

- What is the interest of the (alpha, beta) generalization ? In
  particular in context of neural networks ? And what are the value used
  in the experiments ?
- In the article about RBN, Brooks et al discuss about the amount of
  data need to achieve good performance. Any insight about this for your
  layer ?
- Is there a link between Frechet variance and the variance of the
  Gaussian used for normalization ?

- I guess it should be "neutral element" instead of identity in Eq 13 ?

---

> ### Author Response · Authors · 2023-11-20
> **Response to Reviewer Urfr (R5) (1/2)**
>
> We thank reviewer $\textcolor{purple}{Urfr}$ ($\textcolor{purple}{R5}$) for the encouraging feedback and the constructive comments!😄😄 In the following, we respond to the concerns point by point.
> ***
>
> **1. Table readability.**
>
> Thanks for the suggestive comments. Due to the page limits, we presented the tables concisely. After our work gets accepted, we will move some experimental settings into the appendix and allocate more space for the tables (possibly allowing for two lines or a large figure).
>
> **2. The novelty of the work should emphasized more.**
>
> Thanks for your constructive suggestion! We emphasize our novelty in the introduction in the revised paper (Page 2), highlighting our contributions that our work can theoretically normalize mean and variance and apply to diverse Lie groups.
>
> **3. How to choose hyper-parameters in LieBN.**
>
> Our SPD LieBN has at most three types of hyper-parameters: Riemannian metric, deformation factor $\theta$, and $\mathrm{O}(n)$-invariance parameters $(\alpha,\beta)$. The general order of importance should be Riemannian metric $>$ $\theta$ $>$ $(\alpha,\beta)$.
>
> The most significant parameter is the choice of the Riemannian metric, as all the geometric properties are sourced from the metric. A safe choice would start with AIM, and then decide whether to explore other metrics further. The most important reason is the property of affine invariance of AIM, which is a natural characteristic of covariance matrices. In our experiments, the LieBN-AIM generally achieves the best performance. However, AIM is not always the best metric. As shown in Tab. 4 (b), the best result on the HDM05 dataset is achieved by LCM-based LieBN, which improves the vanilla SPDNet by 11.71\%. Therefore, when choosing Riemannian metrics on SPD manifolds, an appropriate choice would start with AIM and extend to other metrics. Besides, if efficiency is an important factor, one should first consider LCM as it is the most efficient one.
>
> The second one is the deformation factor $\theta$. As we discussed in Sec. 5.1, $\theta$ interpolates between different types of metrics ($\theta=1$ and $\theta \rightarrow0$). Inspired by this, we select $\theta$ around its deformation boundaries (1 and 0). In this paper, we roughly select $\theta$ from $\{ \pm 0.5, \pm 1,\pm 1.5 \}$
>
> The less important parameters are $(\alpha,\beta)$. Recalling Tab. 1, $(\alpha, \beta)$ only affects the Riemannian metric tensor and geodesic distance. For our specific SPD LieBN, they only affect the calculation of variance, which should have fewer effects than the above two parameters. Therefore, we simply set $(\alpha,\beta)=(1,0)$ during experiments.
>
> We have added the above discussion to the App. G.
>
> **4. $\mathrm{O}(n)$-invariance hyper-parameters $(\alpha, \beta)$**
>
> As we stated in the last question above, $(\alpha, \beta)$ is less important than Riemannian metrics and deformation factor $\theta$. Nevertheless, we also conduct experiments on the effect of $(\alpha, \beta)$ here. As $\alpha$ is less important as a scaling factor [1], we set $\alpha=1$ and change the value of $\beta$.
>
> Recalling Eq. (3), $\beta$ controls the relative importance of the trace part against the inner product. Therefore, we set the candidate values of $\beta$ as $\{1,\frac{1}{n},\frac{1}{n^2}, 0, -\frac{1}{n} + \epsilon,-\frac{1}{n^2}\}$, where $n$ is the input dimension of LieBN, and $\epsilon$ is a small positive scalar to ensure $\mathrm{O}(n)$-invariance, *i.e.,* $\min (\alpha, \alpha+n \beta)>0$. $\frac{1}{n^2}$ and $\frac{1}{n}$ means averaging the trace in Eq. (3), while the sign of $\beta$ denotes suppressing (-), enhancing (+), or neutralizing (0) the trace.
>
> We focus on AIM-based LieBN on the HDM05 dataset. We set $\theta=1.5$, as it is the best deformation factor under this scenario. Other network settings remain the same as the main paper. The 10-fold average results are presented in the following table. Note that the dimension $n$ of the input feature of the LieBN layer is 30 in this setting. As expected, $\beta$ has minor effects on our LieBN. We also add this discussion in App. G.1 in our revised paper.
>
> Table 1: The effect of different $\beta$ for AIM-based LieBN on the HDM05 dataset.
> | $\beta$ | -0.0011 | -0.03 | 0.0011 | 0.0333 | 1 | 0 |
> |:-------:|:-------:|:----:|:------:|:------:|:-:|:-:|
> | Mean±STD | 68.18±0.86 | 68.12±0.74 | 68.20±0.85 | 68.18±0.85 | 68.16±0.80 | 68.16±0.68 |

---

> ### Author Response · Authors · 2023-11-20
> **Response to Reviewer Urfr (R5) (2/2)**
>
> **5. Effect of the reduced amount of data.**
>
> Following [2], we use 10% of training data to train SPDNet with and without LieBN on the Radar dataset. Following the main paper, we set $\theta=1, 1, -0.5$ and $(\alpha,\beta)=(1,0)$ for $(\theta,\alpha,\beta)$-AIM, $(\theta,\alpha,\beta)$-LEM, and $(\theta)$-LCM. The 10-fold average results are presented in the table below. We could observe a clear improvement of our LieBN to SPDNet under the limited data availability scenarios.
>
> Table 2: Robustness to lack of data of SPDNet with and without LieBN on the Radar dataset.
> | SPDNet | SPDNetLieBN-AIM | SPDNetLieBN-LCM | SPDNetLieBN-LEM |
> |:------:|:---------------:|:--------------:|:--------------:|
> | 84.17 ± 1.66 | **87.63 ± 1.72** | 86.65 ± 1.48 | 87.60 ± 1.31 |
>
>
> **6. Links between the Fréchet variance and the variance of the Gaussian.**
>
> There are some theoretical links between the Fréchet variance and the variance in the Gaussian distribution in Eq. (12).
>
> Recalling the Gaussian distribution (Eq. (12)) on manifold $\mathcal{M}$
> $$
> p\left(X \mid M, \sigma^2\right)=k(\sigma) \exp \left(-\frac{\mathrm{d}(X, M)^2}{2 \sigma^2}\right).
> $$
> As shown in [3, Props. 4.7], $\mathrm{Var}(X)=\mathbb{E} _{\mathbf{X}}\left[\mathrm{d}^2(X, M)\right]=\sigma^2$. Therefore, $\sigma^2$ is the population variance. The empirical counterpart of population variance is the Fréchet variance [4, P12]. As shown by our Props. 4.2, our LieBN can control the sample variance.
>
> The Maximum Likelihood Estimation (MLE) of $\sigma^2$ does not have a general solution, as $k(\sigma)$ depends on the specific metrics. However, the relation is very clear for the invariant metrics on the Lie groups of SPD manifolds. As shown in [5, Props. 7], under AIM, the Fréchet variance $v ^2$ satisfying $\sigma ^2=\phi(v ^2)$ with $\phi$ strictly increasing. For AIM, our LieBN implicitly controls the population variance through the explicit control of sample variance. For the LEM and LCM, our LieBN can directly transfer the latent Gaussian distribution:
> $$
> \mathcal{N}(M,\sigma^2) \xrightarrow{\text{Centering to }E} \mathcal{N}(E,\sigma^2) \xrightarrow{\text{
> Scaling the variance}} \mathcal{N}(E, s^2) \xrightarrow{\text{Biasing to }B} \mathcal{N}(B, s^2).
> $$
> This result is obtained by Cor. C.4 in App. C and has been discussed in detail in App. C (Page 17).
>
> **7. It should be "neutral element" instead of identity in Eq. (13).**
>
> Thanks for pointing this out! All the "identity" or "identity element" in the paper means neutral element. We agree that the current usage of "identity element" and "identity matrix" might lead to potential confusion. In the original submission, we clarified at the beginning of Sec. 4 that an identity element may not necessarily be an identity matrix. Now we have changed the "identity" element to the "neutral" element throughout the manuscript for better clarity.
>
> > [1] Thanwerdas Y, Pennec X. Is affine-invariance well defined on SPD matrices? A principled continuum of metrics.
> >
> > [2] Brooks D, Schwander O, Barbaresco F, et al. Riemannian batch normalization for SPD neural networks.
> >
> > [3] Chakraborty R, Vemuri B C. Statistics on the Stiefel manifold: theory and applications.
> >
> > [4] Pennec X. Probabilities and statistics on Riemannian manifolds: A geometric approach.
> >
> > [5] Said S, Bombrun L, Berthoumieu Y, et al. Riemannian Gaussian distributions on the space of symmetric positive definite matrices.

---

> > ### Comment · Reviewer_Urfr · 2023-11-22
> >
> > 5) Very interesting ! Thx.
> >
> > 6) Ok.
> >
> > 7) I thinks it's clearer with all these changes.

---

> > > ### Author Response · Authors · 2023-11-22
> > > **Thanks for the instant reply**
> > >
> > > Thanks for the instant reply and suggestive comments!
> > >
> > > For better readability, we have relocated several experimental configurations to the appendix (App. H) and modified all tables in the main paper except Tabs. 2-3.  Notably, we have enlarged Tabs. 4-5, which contain our experimental results, to enhance their readability. Please check our revised paper. 😄😄

---

> > > > ### Comment · Reviewer_Urfr · 2023-11-22
> > > >
> > > > Nice !
> > > >
> > > > For table 2, it is good enough like that (I guess there better ways to present the content, but that is secondary by now).
> > > >
> > > > For table 3, I really like this kind of summary, and although I would deserve a larger place, it is good enough.

---

> > > > > ### Author Response · Authors · 2023-11-22
> > > > > **Thanks for your instant and valuable reply**
> > > > >
> > > > > We have made a minor change on Tab. 2. It is a bit bigger now. Besides, we rearranged the Tab. 3. We think it is much clearer now. 😄

---

> ### Comment · Reviewer_Urfr · 2023-11-22
>
> 1) I am deeply disapointed by this answer: it's the job of authors to deal with the page limit. Promising to (re)move useful parts to the appendix if the paper is accepted is not acceptable.
>
> 2) Nice, it's much better by now.
>
> 3 and 4) Interesting and worth reading, thanks !

---

### Author Response · Authors · 2023-11-17
**Common Response**

We thank all the reviewers for their constructive suggestions and valuable feedback. Below, we address the common questions (CQs).
******

**CQ#1: Difference with previous work [1-3] ($\textcolor{red}{R1}, \textcolor{green}{R3}, \textcolor{blue}{R4}$)**

The main differences are that the RBNs in these three works are either **_designed for a specific manifold or metric_**, or **_fail to control mean and variance_**. In contrast, our LieBN **_performs normalization on general Lie groups_**. We have summarized their limitations in Tab. 2 and discussed the difference in Sec. 3.2. Here, we clarify the difference between our LieBN and their RBNs in detail.

**1. Difference with the RBN proposed in [1]**

**_The RBN in [1] is confined within the standard Affine-Invariant Metric (AIM)_**. In contrast, we implement our LieBN under three invariant metrics: Log-Euclidean Metric (LEM), Log-Cholesky Metric (LCM), and AIM. Moreover, our LieBN further covers the deformed families of LEM, LCM, and AIM.

**_For the specific AIM, our LieBN is also better than the RBN [1] in terms of computational efficiency._** The RBN in [1] is based on parallel transportation, Riemannian exp and log. The specific equation is calculated by the matrix power function (Eq. (10) in [1]). In contrast, our LieBN is based on Lie group left translation, which is fulfilled by Cholesky decomposition (Tab. 3). Note that Cholesky decomposition is more efficient than the matrix power function. As shown in Tab. 5 (b), DSMLieBN-AIM-(1) achieves similar performance to SPDDSMBN with less training time (6.94 v.s. 7.74).

**_Besides, the RBN in [1] is problematic when extending to other manifolds or metrics._** As shown in Eq. (9), the core formulation of the RBN in [1] can be described as
$$
\Gamma _{P \rightarrow Q}(S)=\operatorname{Exp} _Q\left[\mathrm{PT} _{P \rightarrow Q}\left(\operatorname{Log} _P(S)\right)\right].
$$

Since $\Gamma_{P \rightarrow Q}$ becomes affine action under AIM, this formulation can coincidentally control mean under AIM on SPD manifolds. However, the resultant mean and variance are generally agnostic under general manifolds. Therefore, the RBN based on $\Gamma _{P \rightarrow Q}$ could not be extended into other metrics or manifolds. On the contrary, our Props. 4.1-4.2 guarantee the rationality of our LieBN on other Lie groups, such as SO(n). For experiments on SO(n), please refer to the 3rd response to $\textcolor{red}{R1}$.

**2. Difference with the RBN proposed by [3]**

**_Similar to [1], the RBN in [3] generally fails to normalize mean or variance either_,** as this RBN is a variant of $\Gamma_{P \rightarrow Q}$. In contrast, our LieBN can always control both mean and variance on Lie groups.

**3. Difference with the RBN proposed by [4]**

**_The RBN on matrix Lie groups in [4] is also confined within a specific metric and group structures (Sec. 3.2 in [4])_.** On the contrary, our LieBN is designed for general Lie groups. In this sense, the RBN on matrix Lie groups in [4] is a special case of our LieBN. Furthermore, as stated at the end of Sec. 4, our LieBN framework can be extended into right-invariant metrics.

Besides, as stated in Rmk. C.5 in the appendix, the proof presented in [4] is a bit problematic, as the author fails to consider Riemannian volume when dealing with the probability density function. In contrast, our theoretical results in Props. 4.1-4.2 are more solid and general.

**Summary**

In summary, the RBNs in [1-3] are either confined within a specific manifold or metric, or fail to control mean and variance. In contrast, we propose a  framework of batch normalization over general Lie groups and showcase its generality over three families of Lie groups on SPD manifolds.

> [1] Kobler R, Hirayama J, Zhao Q, et al. SPD domain-specific batch normalization to crack interpretable unsupervised domain adaptation in EEG.
>
> [2] Brooks D, Schwander O, Barbaresco F, et al. Riemannian batch normalization for SPD neural networks.
>
> [3] Lou A, Katsman I, Jiang Q, et al. Differentiating through the fréchet mean.
>
> [4] Chakraborty R. Manifoldnorm: Extending normalizations on Riemannian manifolds.

***
For brevity, we refer to reviewers $\textcolor{red}{ZRPd}$ as $\textcolor{red}{R1}$, $\textcolor{brown}{B3vJ}$ as $\textcolor{brown}{R2}$, $\textcolor{green}{LaGM}$ as $\textcolor{green}{R3}$, $\textcolor{blue}{R6Kt}$ as $\textcolor{blue}{R4}$, and $\textcolor{purple}{Urfr}$ as $\textcolor{purple}{R5}$, respectively.

---

### Author Response · Authors · 2023-11-22
**Gentle Reminder (24h Left)**

Dear Reviewers,

Thank you again for the time and effort in reviewing this paper! There are only 24 hours left till the end of the reviewer-author discussion period. Since some of the reviewers have yet to respond or acknowledge reading our response, we are unsure whether our response has fully addressed your concerns.

Please start any conversations if you have any unaddressed concerns. We are happy to answer any further questions. 😄😄

Best regards,

Authors

---

### Meta-Review · Area_Chair_rr8K · 2023-12-06

**Metareview:**

This paper proposed a universal approach to perform batch normalization (BN) on Lie groups. It generalizes existing approaches in the below perspective:  (1) sample statistics (after normalization) are controlled by theory, and (2) applicable to general Lie groups. The algorithm steps are presented. It is tested on SPDNet and shows better performance than existing Riemannian normalization.

*Strengths*:

- A flexible framework based on deep math with guarantees on the sample statistics. New applications of Lie groups should be appreciated as more advanced geometric structures are applied in deep learning.

- The authors carefully positioned their work in the literature, as extensions of Chakraborty and others, and clarified their contribution in comparison tables, with rigorous theoretical developments.

*Weakness*:

- Experimentally, the approach could be tested not only on the three structures of the SPD manifold but also on another type of Lie group (e.g. a toy example), to support the practical usefulness.

- As noted by the authors' rebuttal, the approach has a few hyper-parameters, e.g., the different Lie group structures (corresponding to the choice of a Riemannian metric). A more systematic experimental study on the parameter sensitivity is helpful. The authors' argument on the choice of the Riemannian metrics is based on empirical performance and it is not clear what is the underlying reason that certain metrics are favored.

**Justification For Why Not Higher Score:**

Due to the above-listed weaknesses, this work has space for improvement in its writing and experiments and is not recommended as a top paper. Notably, the empirical study is relatively weak and can be enhanced.

**Justification For Why Not Lower Score:**

This paper advances the SOTA by generalizing existing BN methods with theoretical guarantees. There has been an in-depth discussion in the reviewing process. The authors' normalization is novel, as there are rigorous analytical developments to support it, which is not trivial from previous work (e.g. Chakraborty). The experiments compared three Riemannian metrics of the SPD manifold and showed consistent (although marginal) performance improvement.

Its theory is potentially interesting to the ICLR community.

---

### Decision · Program_Chairs · 2024-01-16

Accept (poster)